# Modeling global radiative effect of brown carbon: A potentially larger heating source in the tropical free troposphere than black carbon

Aoxing Zhang[1], Yuhang Wang[1], Yuzhong Zhang[1,2], Rodney J Weber[1], Yongjia Song[1], Ziming Ke[1,3], Yufei Zou[1,4]

[1]School of Earth and Atmospheric Sciences, Georgia Institute of Technology, Atlanta, USA
[2]Now at School of Engineering and Applied Sciences, Harvard University, Cambridge, MA, USA
[3]Now at Department of Atmospheric Science, University of Wyoming, Laramie, Wyoming, USA
[4]Now at School of Environmental and Forest Sciences, University of Washington, Seattle, WA, USA

*Correspondence to*: Yuhang Wang (yuhang.wang@eas.gatech.edu)

**Abstract.** Carbonaceous aerosols significantly affect global radiative forcing and climate through absorption and scattering of sunlight. Black carbon (BC) and brown carbon (BrC) are light-absorbing carbonaceous aerosols. The direct radiative effect (DRE) of BrC is uncertain. A recent study suggests that BrC absorption is comparable to BC in the upper troposphere over biomass burning regions and that the resulting radiative heating tends to stabilize the atmosphere. Yet current climate models do not include proper physical and chemical treatments of BrC. In this study, we derived a BrC global biomass burning emission inventory on the basis of the Global Fire Emissions Database 4 (GFED4), developed a module to simulate the light absorption of BrC in the Community Atmosphere Model version 5 (CAM5) of Community Earth System Model (CESM) model, and investigated the photo-bleaching effect and convective transport of BrC on the basis of Studies of Emissions, Atmospheric Composition, Clouds and Climate Coupling by Regional Surveys (SEAC$^4$RS) and Deep Convective Clouds and Chemistry Project (DC3) measurements. The model simulations of BC were also evaluated using HIAPER (High-Performance Instrumented Airborne Platform for Environmental Research) Pole-to-Pole Observations (HIPPO) measurements. We found that globally BrC is a significant absorber, the DRE of which is 0.10 W/m$^2$, more than 25% of BC DRE (+0.39 W/m$^2$). Most significantly, model results indicated that BrC atmospheric heating in the tropical mid and upper troposphere is larger than that of BC. The source of tropical BrC is mainly from wildfires, which are more prevalent in the tropical regions than higher latitudes and release much more BrC relative to BC than industrial sources. While BC atmospheric heating is skewed towards northern mid-latitude lower atmosphere, BrC heating is more centered in the tropical free troposphere. A possible mechanism for the enhanced convective transport of BrC is that hydrophobic high molecular weight BrC becomes a larger fraction of the BrC and less easily activated in a cloud as the aerosol ages. The contribution of BrC heating to the Hadley circulation and latitudinal expansion of the tropics is likely comparable to BC heating.

# 1 Introduction

Carbonaceous aerosols, including black carbon (BC) and organic carbon (OC), are important factors in global atmospheric radiative forcing. BC warms the atmosphere by directly absorbing solar radiation (Bond et al., 2013). OC used to be thought to cool the atmosphere due to its light scattering properties. However, some OC, known as "brown carbon" (BrC), absorbs visible light with a wavelength dependence; the efficiency increases rapidly with decreasing wavelength (Hecobian et al., 2010; Kirchstetter and Thatcher, 2012; Kirchstetter et al., 2004; Yang et al., 2009).

The primary source of BrC is incomplete combustion of biomass and biofuel (Chakrabarty et al., 2010; Feng et al., 2013; Desyaterik et al., 2013; Washenfelder et al., 2015). There is evidence that BrC is also associated with fossil fuel combustion and urban emissions (Zhang et al., 2011; Costabile et al., 2017; Yan et al., 2017; Xie et al., 2017). Secondary BrC is mainly produced from the photo-oxidation of volatile organic compounds (VOCs), such as nitrophenols and aromatic carbonyls (Jacobson, 1999; Nakayama et al., 2010; Nakayama et al., 2013), monoterpenes (Laskin et al., 2014), and methylglyoxal (Sareen et al., 2013). Secondary BrC also comes from aqueous-phase reactions in droplets (Updyke et al., 2012; Nguyen et al., 2012) and homogenous and heterogenous reactions of catechol (Pillar et al., 2014; Pillar and Guzman, 2017; Magalhães et al., 2017) and phenolic compounds (Yu et al., 2016; Lavi et al., 2017; Smith et al., 2016). BrC from biomass burning contributes more to light absorption than the other sources (Chakrabarty et al., 2010; Saleh et al., 2014; Kirchstetter and Thatcher, 2012; McMeeking, 2008). Factor analysis of water-soluble organic carbon (WSOC) over the southeastern United States averaged for one year (Hecobian et al., 2010) attributed ~50% of solar absorption at 365 nm to biomass burning emissions, 20-30% to secondary organic carbon, and ~10% to primary urban emissions (mobile sources).

Alexander et al. (2008) analyzed the radiative effects of aerosols in the outflow region of East Asia and found that wood smoke BrC accounted for 14% of total aerosol absorption. Liu et al. (2014) found a ~20% reduction of aerosol cooling from BrC absorption at the top of the atmosphere on the basis of measured BrC vertical profiles. However, current observations do not provide enough constraints on the global BrC radiative forcing (Schuster et al., 2016a; Schuster et al., 2016b).

Global models have been applied to estimate Direct Radiative Forcing (DRF) and Direct Radiative Effects (DRE) of BrC. Aerosol DRE represents the difference of radiative budget with and without aerosols, and DRF represents the difference of DRE between present day and pre-industrial times (Heald et al., 2014). The study by Feng et al. (2013) suggested a +0.04-0.11 $W/m^2$ warming effect at the top of atmosphere due to the absorption of BrC, and attributed 19% of anthropogenic aerosol absorption to BrC. Wang et al. (2014) estimated the global DRF of +0.11 and +0.21 $W/m^2$ for BrC and BC, respectively. Jo et al. (2016) estimated a BrC DRE of +0.11 $W/m^2$. Lin et al. (2014) estimated a BrC DRE of +0.22-0.57 $W/m^2$, which accounted for 27%-70% of the BC absorption in their model. Park et al. (2010) modeled BrC over East Asia and calculated a DRE of +0.05 $W/m^2$ at the top of the atmosphere. Brown et al. (2018) estimated a BrC DRE of +0.13±0.01

W/m$^2$ and 0.01± 0.04 W/m$^2$ from BrC aerosol-cloud interaction. Saleh et al. (2015) estimated a BrC DRE of 0.22 W/m$^2$ when BrC externally mixed with BC, and 0.12 W/m$^2$ when BrC is internally mixed with BC. All of these model estimations of BrC DRE and DRF treated BrC similar to BC, where properties were invariant with atmospheric processing or aging.

Laboratory and field studies showed a reduction of BrC absorption when exposed to light, which is usually referred to as
"photo-bleaching" (Zhao et al., 2015). A recently global model simulation (Wang et al., 2018) included this effect, constrained with BrC absorption measurements in DC3 and SEAC$^4$RS, resulting in a large reduction of global BrC DRE to +0.048 W/m$^2$ compared to previous estimates from the studies listed above. Brown et al. (2018) developed a BrC module in the CESM and showed a reduction of BrC DRE to 0.06±0.008 W/m$^2$ because of photo-bleaching. Other effects of atmospheric processing have not yet been in BrC global modeling. Results from the Deep Convective Clouds and Chemistry
Project (DC3) found high concentrations of BrC in the continental upper tropospheric due to convective transport, suggesting more efficient atmospheric vertical transport of BrC than previously assumed (Zhang et al., 2017). In this study, we developed and implemented a BrC module in the Community Earth System Model (CESM) to assess the effects of BrC DRE. Here, we include these effects and make use of the aircraft measurements of BrC absorption profiles from DC3 and SEAC$^4$RS campaigns to evaluate the model formulation and simulation results. The global BrC emissions from biomass
burning, biofuel emissions, and secondary formation were included. We tested the sensitivity of photo-bleaching effect and the deep convective transport of BrC to its DRE, and estimated the global DRE. Model simulation results without considering the differential convective transport and BC and BrC are compared to previous studies. This is the first attempt to comprehensively analyze how convective transport and photo-bleaching affect global atmospheric heating by BrC absorption relative to BC.

## 2 Model Description

### 2.1 The CESM Model

We developed the brown carbon simulation based on the Community Earth System Model (CESM) version 1.2.2 and its atmospheric component, the Community Atmosphere Model version 5 (CAM5) (Neale et al., 2010). The CAM5 model has a comprehensive mechanism for aerosols and cloud-aerosol interaction (Liu et al., 2012; Ghan et al., 2012; Gettelman et al.,
2010). The CAM5 radiation scheme is the Rapid Radiative Transfer Method for GCMs (RRTMG), which provides an accurate method for radiative flux calculation (Iacono et al., 2008; Mlawer et al., 1997). The atmospheric chemistry module, turbulence scheme, convection scheme, and cloud physics are coupled in the model (Zhang and McFarlane, 1995; Bretherton and Park, 2009; Park and Bretherton, 2009; Richter and Rasch, 2008; Morrison and Gettelman, 2008). We used a 3-mode version of the modal aerosol model (MAM3) for aerosol modeling in CAM5. The 3 modes are Aitken, accumulation, and
coarse modes (Neale et al., 2010). In our BrC simulations, we used CAM5 with a spatial resolution of 1.9°x2.5°. The wet scavenging scheme of aerosols in CAM5 includes below-cloud scavenging and in-cloud scavenging, which was found to

have a high bias (Wang et al., 2011; Liu et al., 2012). For the simulations used to compare with field observations, we nudged CAM5 meteorological field (temperature, humidity, wind, surface pressure and heat) to the same meteorological year, month, and day as the observations using GEOS-5.2.0 meteorological data products (Suarez et al., 2008) every 6 hours in order to evaluate the model simulations with BrC observations (Ma et al., 2013; Chipperfield, 2006). We also conducted 5-year free-running model simulations using the climatology of 2010 to analyze the climate response to BrC and BC heating. The spin-up time is 3 months in the nudged CAM5 simulations and is 1 year in the free-running simulations.

## 2.2 Emissions

We derived global BrC emissions from biomass burning, biofuel, and secondary formation. Same as Brown et al., 2018, we used the parameterization by Saleh et al. (2014) for biomass and biofuel burning, in which emitted BrC absorption is a function of BC/Organic Aerosols (OA) emission ratio. We included secondary aerosols from the oxidation of aromatic as the major source of secondary BrC (Hecobian et al., 2010; Sareen et al., 2013; Lin et al., 2015). Secondary BrC produced from aromatic oxidation absorbs more solar radiation in high-NOx conditions (Laskin et al., 2015; Lin et al., 2015; Liu et al., 2012; Nakayama et al., 2010, 2013; Yu et al., 2014; Zhong and Jang, 2011). We did not consider the NOx dependence of secondary BrC in this study. More details will be described in section 3.

The biomass burning emissions we used are from the Global Fire Emission Database version 4 including small fires (GFED4s) (Giglio et al., 2013; Randerson et al., 2012). It contains global burned area distribution and biomass burning emission factors of related aerosols and gas species for different fire types, with a spatial resolution of 0.25°x0.25°. In CAM5, we aggregated it to a spatial resolution to 1.9°x2.5° and used GFED daily emission and diurnal cycle factors. The different emission factors for tropical forest, temperate forest, boreal forest, savanna, agriculture waste and peat burning are based on Akagi et al. (2011). Fire emissions can reach high altitudes (e.g., Neale et al., 2010). We used an observation-constrained global fire plume rise dataset in which MODIS fire hotspot and fire radiance power data were used in a 1-D fire plume rise model and the resulting fire plume distribution is in good agreement with the Multi-angle Imaging SpectroRadiometer (MISR) observations (Ke, 2019). Biomass burning emissions have high uncertainties caused by burned area, emission factors, fuel loads and combustion completeness factors (Akagi et al., 2011; Giglio et al., 2013), and the complex interactions between fire, terrestrial ecosystem, and climate systems amplify these uncertainties (Zou et al., 2019).

Anthropogenic emissions are from the IPCC AR5 emission dataset (Lamarque et al., 2010), and BC and OC emissions are updated based on the emission inventory of 2000 (Bond et al., 2007; Junker and Liousse, 2008). We increased anthropogenic emissions in China by 50% according to Zhang et al. (2009). For the Arctic region (> 66°N), we used the Evaluating the

Climate and Air Quality Impacts of Short-lived Pollutants (ECLIPSE) emission dataset, which includes an improvement for the Arctic BC emissions (Stohl et al., 2013; Klimont et al., 2015).

For the optical properties of BC, we used 10 and 8.1 m2/g for 345- 442 nm and 442-625 nm, respectively, as the MAE values of BC (Knox et al., 2009; Bond and Bergstrom, 2006). The MAE values are lower than the estimation by Bond et al. (2013) (11 $m^2$/g) and Jacobson (2016) (16 $m^2$/g including high-RH conditions), and are higher than the estimation by Schulz et al. (2006) (7.9±1.9 $m^2$/g). MAM3 assumes that primary carbon is internally mixed with secondary aerosols in the accumulation mode.

## 3 Brown Carbon Module

### 3.1 BrC optical property and photo-bleaching

BrC absorption depends on its Mass Absorption Efficiency (MAE), which is the ratio of light absorption in the wavelength $\lambda$ to BrC mass concentration ($m^2$ $g^{-1}$):

$$MAE_{BrC}(\lambda) = \frac{A(\lambda)}{c_{BrC}} . \tag{1}$$

where $A(\lambda)$ represents the absorption of BrC at the wavelength of $\lambda$ ($m^{-1}$), and $c_{BrC}$ is the mass concentration of BrC (g $m^{-3}$).

Similar to Jo et al. (2016), we used a constant MAE value for primary BrC, 1.0 $m^2$/g at 550 nm (McMeeking, 2008), and we used an MAE value of 0.19 $m^2$/g at 550 nm for secondary BrC based on the work by Nakayama et al. (2010). There are other MAE estimates in experiments such as 3.6-4.1 $m^2$/g (Alexander et al., 2008) and 0.58-0.64 $m^2$/g (Hecobian et al., 2010), and model-specified values such as 0.35 $m^2$/g by Wang et al. (2018). MAE value may also change in different seasons (Cheng et al., 2011). At present, there are not enough observations to specify variable MAE values in the model. The MAE value at the other wavelengths was calculated using the following function (Bond and Bergstrom, 2006):

$$MAE(\lambda) = MAE(\lambda_0) * (\lambda_0/\lambda)^{AAE}, \tag{2}$$

where AAE is absorption Angström exponent, and $\lambda_0$ is 550 and 365 nm for primary and secondary BrC, respectively. We used AAE = 5.0 for $\lambda$ < 2 µm (Jo et al., 2016; Kirchstetter and Thatcher, 2012). BrC AAE varies depending on its source and wavelength used (Kirchstetter and Thatcher, 2012; Liu et al., 2014). Jo et al. (2016) found BrC/BC ratio decreases when the BrC AAE increases from 5 to 6.19, Saleh et al. (2014) also found BC/OA ratio is negatively related to BrC AAE and positively related to BrC absorption. Therefore, the variation of BrC AAE leads to additional uncertainty of the BrC simulation.). The imaginary part of refractive index for BrC is estimated using the following equation (Liu et al., 2013):

$$k_{BrC,\lambda} = k_{OA,\lambda} * \frac{c_{OA}}{c_{BrC}} = \frac{\rho\lambda \cdot A(\lambda)}{4\pi \cdot c_{BrC}} = \frac{\rho\lambda \cdot MAE(\lambda)}{4\pi}, \tag{3}$$

where $\rho$ is particle density (g $m^{-3}$), A($\lambda$) is the light absorption at wavelength $\lambda$, and c is mass concentration.

The estimated $k_{BrC}$ value is 0.045 at 550 nm for primary BrC and 0.043 at 365 nm for secondary BrC, respectively.

There is observational evidence that both primary and secondary BrC are affected by photochemical aging (or bleaching), which reduces BrC light absorption when exposed to light (Forrister et al., 2015; Sareen et al., 2013; Lee et al., 2014; Zhong and Jang, 2011; Wong et al., 2017; Wong et al, 2019). Previous modeling of the BrC photo-bleaching effect by Wang et al. (2018) and Brown et al. (2018) applied a 1-day e-folding time for BrC before reaching a threshold of 25% of the original BrC absorption. Our approach to BrC photo-bleaching considers different bleaching effects depending on BrC source. We specify a decay half-life of 12 hours when light is present for primary biomass and biofuel BrC in the daytime until 6% is left and no further photobleaching occurs (Forrister et al., 2015) due to stable high molecular weight chromophores (Di Lorenzo and Young, 2015; Di Lorenzo et al., 2017; Wong et al., 2017; Wong et al, 2019). Different components of SOA have different photo-bleaching lifetimes. Aromatic SOA has a half-life of 12-24 hours (Liu et al., 2016; Lee et al., 2014; Zhong and Jang, 2011), limonene SOA has a half-life of <0.5 hours (Lee et al., 2014). Methylglyoxal SOA has a half-life of 90 minutes (Zhao et al., 2015; Wong et al., 2017). Therefore, the half-life for secondary aromatic BrC is specified at 12 hours in daytime until it is completely removed (Liu et al., 2016). The other secondary BrCs that have shorter lifetimes contribute little to global radiative forcing and are not included in the model.

The analysis of aircraft BrC observations by Zhang et al. (2017) showed that BrC transported by deep convection plays a significant role in radiative heating of the upper troposphere, and that BrC warming is about one third of BC warming at the tropopause. The standard model simulations show a large low bias of BrC in the upper troposphere compared to the observations by Zhang et al. (2017). In addition, in-cloud heterogeneous BrC production is another possible reason for the high level BrC in the upper troposphere. A fraction of biomass burning BrC from heterogeneous oxidation by ozone will stay free from photo-bleaching, and BrC from heterogeneous oxidations by OH has a long lifetime of days (Browne et al., 2019). Therefore, we conducted sensitivity simulations of BrC to investigate the effects of photo-bleaching and wet scavenging during deep convection.

One important finding by Zhang et al. (2017) is that wet scavenging of BrC during convection differs from BC and OC. Therefore, BrC is simulated using a different tracer from OC in this work unlike Brown et al. (2018). The BrC property of interest is absorption and we assume that the tracer's optical property is light absorption only (no scattering). Consequently, there is no double counting of OC scattering. However, it should be noted that BrC is a class of organic aerosols that both scatter and absorb light. We analyzed in this study the effect of BrC light absorption. The model simulation of OC mass and scattering was not affected by the simulation of a BrC tracer that only absorbs light. In the following analysis, the DRE from BrC is for light absorption only such that it represents the DRE of the OC absorption and can be compared to the DRE of the BC absorption.

### 3.2 BrC Emissions

We assumed that BrC is emitted in the accumulation mode (Liu et al, 2013). Sources of the more stable forms of BrC include primary emissions of biomass burning and biofuel, as well as secondary formation from aromatic oxidation. Similar to previous model approaches (Wang et al., 2018; Brown et al., 2018), biomass burning BrC emissions were parameterized as a function of BC to OA ratio of the emissions (Saleh et al., 2014):

$$k_{OA,550} = 0.016 log_{10}\left(\frac{E_{BC}}{E_{OA}}\right) + 0.03925, \tag{4}$$

where $k_{OA,550}$ is OA absorptivity at 550 nm, and $E_{BC}$ and $E_{OA}$ are BC and OA emission rates (g m$^{-2}$ s$^{-1}$), respectively. We computed $k_{OA,550}$ in order to calculate BrC emissions. In the model, the absorption of the OC tracer was specified to be 0. All OC absorption was due to the BrC tracer. We scaled BrC Emissions based on $k_{OA,550}$, MAE and OA emissions using the following equation by Liu et al. (2013):

$$E_{BrC} = \frac{4\pi k_{OA,550} \cdot E_{OA}}{\rho \cdot 550nm \cdot MAE_{BrC}(550\,nm)}. \tag{5}$$

where $\rho$ is particle density (g m$^{-3}$), and $E_{BrC}$ is BrC emission rate (g m$^{-2}$ s$^{-1}$).

Using the GFED emissions inventory, we estimated an annual average global BrC source from biomass burning of 3.6 TgC/yr, ~23% of OC emissions (15.9 TgC/yr) and about twice as large as BC emissions (1.9 TgC/yr). The variability of BrC emission rate among biomes therefore depends on the BC to OA emission ratios in the GFED emission inventory. Using the same equations and an average $E_{BC}/E_{OA}$ ratio of 0.41 (Junker and Liousse, 2008), we estimated an $E_{BrC}/E_{OA}$ ratio of 0.38 and an annual global BrC biofuel source of 3.1 TgC/yr on the basis of biofuel emission inventory by Bond et al. (2007). The estimates of primary BrC emissions are comparable to previous studies (Table 1). BrC emissions from fossil fuel combustion are not yet well characterized to be included in a global model (Saleh et al., 2014; Xie et al., 2017).

The major fraction of secondary BrC that affects atmospheric heating is formed during the oxidation of aromatics (Jacobson, 1999; Nakayama et al., 2013; Zhong et al., 2012). As in previous studies (Jo et al., 2016; Wang et al., 2014), we assumed that secondary BrC is from aged aromatic SOA. In the CAM5 MAM3 aerosol mechanism, the SOA mass yield for aromatics oxidation is 15% (Neale et al., 2010; Odum et al., 1997). We estimated a global secondary BrC source of 4.1 TgC/yr in agreement with previous studies.

BrC emissions used in this study and the comparison to previous studies are summarized in Table 1, and the emission distribution is shown in Figure 1. Biofuel emissions are high in China and India, and secondary BrC sources are also large in Europe and North America. Figure 2 shows the annual cycle of BrC emissions in 2010. Biofuel and secondary BrC sources have little seasonal variation in the model, while biomass burning has significant seasonal variation. The BrC source is the highest in August at 18.9 TgC/yr. Biomass burning emission accounts for more than 60% of the BrC emissions in August.

**4 Model evaluation**

**4.1 Black carbon measurements from HIPPO**

HIAPER (High-Performance Instrumented Airborne Platform for Environmental Research) Pole-to-Pole Observations (HIPPO) measured atmospheric composition approximately from the Arctic to the Antarctic over the Pacific Ocean (Wofsy, 2011). HIPPO executed 5 missions from January 2009 to September 2011. The flight path of each HIPPO mission is shown in Figure 3. Measurements over continental North America east of 140°W were not included in our model evaluation. BC measurements for particles with a size range of 90-600 nm were made from a single-particle soot photometer (SP2) and we increased measured BC data by a factor of 1.1 to account for larger sized BC in the model evaluation (Schwarz et al., 2010; Schwarz et al., 2013). We make use of HIPPO BC measurements to constrain convective transport and wet scavenging.

Wet scavenging is uncertain in 3-D global modelling of BC (Schwarz et al., 2010; Liu et al., 2011). Wang et al. (2013) tested the sensitivities of different physical mechanisms and found a high sensitivity of BC simulations to convective transport and wet removal. Comparison of CAM5 BC simulations with HIPPO observations in Figure 4 shows large overestimates of BC in the tropics and the upper troposphere. Since the emissions of BC are from the surface, the model high biases in these regions suggest insufficient wet scavenging during convection. Wang et al. (2014) updated the model wet scavenging by scavenging hydrophobic aerosols in convective updrafts and scavenging hydrophilic aerosols from cold clouds. In all simulations of this study, we increased interstitial BC scavenging by a factor of 5 to increase wet scavenging and reduced stratiform liquid-containing cloud based on model evaluations using HIPPO observations. The high biases above 300 hPa at mid and high latitudes persisted particularly for HIPPO-1 in January 2009 and HIPPO-2 in November 2009. In winter, the simulated high BC concentrations were above the tropopause level at mid and high latitudes, indicating that convective transport reached too high an altitude. We therefore limited deep convection mass transport to an altitude of 50 hPa below the model estimated tropopause, after which the high biases at mid and high latitudes above 200 hPa were corrected. During HIPPO-3 in March-April 2010, model simulated free tropospheric BC at northern mid and high latitudes were much lower than the observations, suggesting excessive scavenging in the model. We reduced cloud-born BC scavenging to 10% when cloud ice is present and to 50% for the other conditions, which improved the model simulations of free tropospheric BC simulations at mid and high latitudes in HIPPO-3 (and HIPPO-2). The modification slightly worsened the model high bias at northern mid and high latitudes in the summer for HIPPO-5 in August-September 2011 and to a lesser extent for HIPPO-4 in June-July, 2011. Figure 5 shows the comparison between BC vertical profiles during all HIPPO campaigns with CAM5 simulations for 5 latitude bins (90°S-60°S, 60°S-20°S, 20°S-20°N, 20°N-60°N, 60°N-90°N), respectively. The modified CAM5 simulations agreed better with the observations in all regions, but still overestimated BC in the middle and upper troposphere over the tropics, which may lead to a low bias in the model simulated BrC/BC heating ratio in the tropics (to be discussed in section 5.3).

**4.2 Aerosol optical depth (AOD) and absorption aerosol optical depth (AAOD) over fire emission dominated regions**

Direct assessments of BrC sources using observations are difficult because of limited observations. We can, however, evaluate model simulations of fire aerosols with AOD and AAOD measurements. For this purpose, we chose the months and regions in model simulations that >50% of monthly mean AOD data are from fire emissions for 2010. The distribution of model simulated mean AOD for data points selected in this manner are shown in Figure S1(a). For comparison purpose, the ground-based AOD measurements were obtained from the Aerosol Robotic Network (AERONET) version 3 level 2.0 dataset (Holben et al., 1998). To compare with model simulated AOD data at 550 nm, the AOD measurements at 500 and 675 nm were used to compute Ångström exponent (Ångström, 1964) and calculate the corresponding AOD values at 550 nm (Kumar et al., 2013). Figure 6(a) compares the monthly mean fire dominated 550 nm AOD observations in 2010 in AERONET with corresponding monthly mean model results for selected months and regions. The correlation coefficient, $r$, is high at 0.88. We performed principal-component (PC) regression analysis of observed and simulated data. The low regression slope (0.56) indicates that the observed AOD data were underestimated, implying a low bias in fire emissions.

We also compared AOD with the measurements from Moderate Resolution Imaging Spectroradiometer (MODIS) on Terra (EOS-AM-1) satellite for the months and regions in model simulations that >50% of monthly mean AOD data are from fire emissions for 2010. We used Collection 6 of MODIS level-3 Deep Blue/Dark Target merged product with a resolution of 1°x1° (Platnick et al., 2017). Figure 6(b) shows the comparison. Both the correlation coefficient $r$ (0.67) and the PC regression slope $k$ (0.49) are lower than the comparison with AERONET observations. One reason is that the high AOD data in the outflow region of the tropical Atlantic from fire emissions over Africa were significantly underestimated (Figure S1(b)); similar low biases were also found in the outflow region of fire emissions in South America. Additionally, CAM5 underestimates AOD at high latitudes (Liu et al., 2012). The general low bias of fire aerosol emissions was also found by Ward et. al. (2012). For these data, the effect of BrC absorption on AOD is small; we estimate that BrC absorption contributes 0.37% of the total AOD.

In addition, we compared the model simulations to the AAOD data from the AERONET version 3 level 2.0 inversion dataset (Holben et al., 2006). Since the AAOD estimation is highly uncertain in the low AOD conditions (Dubovik et al., 2000), we used only AAOD measurements for AOD at 440 nm $\geq$ 0.4 (Holben et al., 2006). Monthly mean AAOD data were computed for AERONET sites with more than 10 days of daily averaged observed AOD at 440 nm > 0.4 in a month. Because of the model underestimation, the corresponding model threshold of AOD at 440 nm is 0.315 based on the PC regression between AERONET observations and model simulation results (Figure 6(a)). Daily model results with AOD at 440 nm > 0.315 were used to compute simulated monthly means for the grid cells corresponding to the AERONET sites. To show the performance of the model simulation of aerosol absorption, here we compare the observed and simulated AAOD values at 550 nm, which is near the peak wavelength of solar intensity. Because of the strong wavelength dependence of BrC absorption, the

enhancement of AAOD by BrC absorption at wavelengths lower than 550 nm is more significant. Figure 6(c) compares the monthly mean 2005-2014 AERONET AAOD data over fire-dominated regions and months with the corresponding monthly mean model results. The observations showed significant interannual variability, which was not included in the model results for a climatological 2010 year. With BrC absorption, the simulated higher AAOD data are in better agreement with AERONET observations with a PC regression slope of 0.59 compared to a slope of 0.43 for the simulation without BrC absorption. For these observations, the model underestimated the AERONET AAOD observations by 39% without BrC absorption. Including BrC absorption reduced the low bias to 17%, which is well within the large variability of the observations. Globally, the AAOD absorption at 550 nm is higher by 8.5% on average when BrC is considered in the model simulations.

## 5 Results

### 5.1 Model simulations of BrC for DC3 and SEAC[4]RS missions

We evaluated BrC model simulations using the measured BrC absorption data from the airborne measurements of Studies of Emissions, Atmospheric Composition, Clouds and Climate Coupling by Regional Surveys (SEAC[4]RS) (Toon et al., 2016) and Deep Convective Clouds and Chemistry Project (DC3) field experiments (Barth et al., 2015). The SEAC[4]RS campaign was conducted during August 6 to September 23, 2013 over the central and southeast U.S., and the DC3 campaign was conducted from May 18 to June 22, 2012 over a similar region. Flight tracks for these experiments were shown in Figure 7. Fresh fire plume data, diagnosed by plumes with a coefficient of determination between CO and $CH_3CN > 0.5$ during the period of enhanced CO, were not included in model evaluation as in previous studies (De Gouw et al., 2004; Liu et al., 2014).

We described in section 3.1 the rationale for sensitivity simulations to evaluate the effects of BrC photo-bleaching and convective wet scavenging. The model sensitivity simulations are listed in Table 2. In the NCNB (base) model, neither effect was included. In the NCB model, the photo-bleaching effect is included. In the ICNB model, the wet scavenging efficiency of convective transported BrC was decreased from 75% simulated in the base model to 30%, such that ~70% of BrC was transported through convection to the free troposphere as suggested by Zhang et al. (2016). In the ICB model, both photo-bleaching and reduced convective scavenging effects were included. The ICBB model is similar to ICB model, but photo-bleaching of all BrC was included; in the other models including the photo-bleaching effect, only non-convectively transported BrC was affected (Zhang et al., 2017).

Figure 8 shows the observed vertical profiles of BrC absorption, BrC to BC absorption ratio (BrC/BC ratio) and concentrations of BC and CO during the DC3 and SEAC[4]RS experiments in comparison to the corresponding model

simulation results. The difference between BC and CO vertical profiles is negligible among the sensitivity simulations. Simulated mean BC concentrations are within the uncertainties of the measurements. The underestimation at 2-5 km during SEAC[4]RS likely reflects underestimated fire emissions since the coefficient of determination ($R^2$) is 0.6 for HCN and BC at 2-5 km and it is 0.5 for HCN and BrC, reflecting the effects of biomass burning emissions on BC and BrC. The higher CO concentrations in the model than the observations, particularly near the surface, suggests that the model overestimates surface CO emissions.

Table 2 lists all sensitivity simulations. For BrC and BrC/BC simulations, Figure 8 shows that the NCNB model clearly overestimated BrC compared to the observations at 0-8 km (the overestimate is not as apparent in the BrC/BC comparison because it is a logarithmic scale). The overestimation reflected the importance of photo-bleaching (Forrister et al., 2015; Sareen et al., 2013; Lee et al., 2014; Wang et al., 2016; Wong et al., 2017; Wong et al, 2019; Zhong and Jang, 2011). The overestimation in the lower troposphere in NCNB led to a reasonable simulation of BrC in the upper troposphere, although the underestimation at 12 km was obvious relative to the ICB simulation during the DC3 experiment. Similarly, considering enhanced convective transport, but not photo-bleaching, the ICNB simulation clearly overestimated BrC absorption relative to the observations. Including photo-bleaching, but not enhanced convective transport of BrC, the NCB simulation clearly underestimated BrC and the BrC/BC ratio in comparison to the observations. We also included a simulation of ICBB, in which enhanced convective transport of BrC was included with photo-bleaching. Compared to the observations, upper tropospheric BrC and the BrC/BC ratio in the ICBB simulation were clearly underestimated. At 12 km, the observed BrC/BC ratio is ~10 and ~20 times higher than BrC/BC near the surface during DC-3 and SEAC4RS, respectively. This increase in the BrC/BC ratio in the upper troposphere was captured by the ICB simulation. On the basis of our current understanding of BrC processes (Forrister et al., 2015; Sareen et al., 2013; Lee et al., 2014; Wang et al., 2016; Wong et al., 2017; Wong et al, 2019; Zhang et al., 2017; Zhong and Jang, 2011) and the model evaluation with the observations, we chose the ICB simulation to investigate the effects of global BrC radiative forcing.

During the SEAC[4]RS experiment, Figures 8 shows that the models overestimated both BC and BrC in the upper troposphere (except ICBB and NCB, which underestimated BrC in both experiments), and all model simulations except ICNB was underestimated the BrC/BC ratio in the upper troposphere and all model simulations except NCB was overestimated the BrC/BC ratio in the middle-lower troposphere. The simulation bias is mostly due to biases in BC simulation in that BC was overestimated in the upper troposphere and underestimated in the middle-lower troposphere (Figure 8).

In the ICB simulation, wet scavenging of BrC was reduced relative to BC in order to simulate the observed BrC/BC ratios in DC3 and SEAC[4]RS. The mechanisms are not yet clear due to a lack of laboratory and field observations. Hydrophobic OC, such as humic-like substances (HULIS), is more likely to have high light-absorption compared to hydrophilic OC (Hoffer et al., 2006). BrC with high molecular weight dominates the aged biomass burning plume (Wong et al., 2017, Wong et al.,

2019). Since higher molecular weight compounds have lower hygroscopicity (Dinar et al., 2007), and it is harder to activate hydrophobic OC in cloud, less BrC is removed in deep convection. Another possible mechanism is production of BrC through in-cloud heterogeneous processing of fire plumes (Zhang et al., 2017). However, there is no observation data to implement such a mechanism in a model.

## 5.2 Simulated global zonal mean distribution of BrC

We performed diagnostic model simulations to investigate the contributions of BrC absorption from biomass burning emissions, biofuel emissions, and secondary formation, respectively. Figure 9 shows the results. Secondary BrC production has a relatively small contribution because of photo-bleaching of secondary BrC is 100%, while a small fraction of BrC is left after photo-bleaching of biomass burning and biofuel BrC (Forrister et al., 2015). Both biofuel and secondary production are largest at northern mid latitudes since they are due to anthropogenic emissions.

Biomass burning BrC shows drastically different distributions from biofuel BrC. The latitudinal maximum is in the tropics and subtropics, with a secondary peak at 60°N due to fires over Canada and Siberia. The vertical extent of biomass burning BrC is much higher than biofuel BrC due to fire plume rise (Ke, 2019) and the higher vertical extent of tropical convection than midlatitudes. While the effect of biofuel BrC is primarily in the lower troposphere, the radiative forcing of biomass burning BrC is much more substantial in the free troposphere and therefore more strongly affects the atmosphere since solar heating of the atmosphere is generally weak.

## 5.3 Global directive radiative effect of BrC

Aerosol DRE represents the instantaneous radiative effect of aerosols, which is sometimes confused with DRF (Heald et al., 2014, Ghan, 2013). We applied the Rapid Radiative Transfer Method for GCMs (RRTMG) to BrC and BC radiative forcing. We parameterized the imaginary part of BrC refractive index as an external input of RRTMG. As discussed in Section 3.2, the imaginary refractive index is specified at 0.045 at 550 nm and 0.043 at 365 nm for primary and secondary BrC, respectively. The wavelength boundaries for RRTMG shortwave and longwave are listed in Tables S1a and S1b in the supplement (Neale et al., 2010; Iacono et al., 2008; Mlawer et al., 1997). We calculated the imaginary refractive index at a different wavelength by introducing wavelength dependence $w$ (Saleh et al., 2014):

$$w = AAE - 1, \tag{6}$$

$$k_{BrC,\lambda} = k_{BrC,550} \times \left(\frac{550}{\lambda}\right)^w, \tag{7}$$

where $k_{BrC,\lambda}$ denotes the imaginary refractive index of BrC, and $w$ is the wave-length dependent AAE value. Calculation and parameterization of MAE and the imaginary refraction index of BrC were discussed in Section 3.1.

In our estimation of BrC DRE, we only considered the absorption of BrC, and the effect of scattering is not considered. We computed the clear sky net solar flux at the top of atmosphere in two simulation, one with BrC tracer absorbing light and the other without. The difference between the two simulations is BrC DRE. The same method was used to calculate BC DRE.

The ICB model calculated global DRE distributions of BrC and BC and the DRE ratio of BrC to BC are shown in Figure 10. For 2010, we estimated the global averaged DRE of BrC absorption at 0.10 $W/m^2$ in comparison to 0.39 $W/m^2$ by BC. While the global DRE by BrC is less than BC, regional BrC DRE can be as large as that of BC due to the large difference in emission distributions. BC DRE is large at northern mid latitudes due to anthropogenic emissions from China and India. BrC

emissions are relatively low in these regions (Figure 1) and consequently BC DRE dominates. Over regions with large fire emissions, both BC and BrC are important. Over most regions of the remote tropical ocean, BrC DRE is larger than BC, despite a model high bias of simulated BC in the middle and upper tropical troposphere (Figure 5), suggesting significant broad regional effects by BrC radiative forcing in the tropics. This simulated feature is due to two factors. In the ICB simulation, wet scavenging removes much more BC than BrC. Therefore, BrC is enriched relative to BC in the free

troposphere (e.g., Zhang et. al., 2017). In the tropics, the easterly trade wind in the boundary layer become westerlies in the middle and upper troposphere. The regions of boundary layer BC transport and free tropospheric BrC transport are in opposite directions. As a result, the DRE ratio of BrC to BC is low to the east of the fire emission regions and it is high to the west of the fire emission regions.

To discuss our simulation results in the context of previous modeling studies, which did not use the ICB assumptions, we show the annual mean DRE distributions for all model simulations (Table 2) in Figure 11. Comparing the global mean DRE relative change of ICB (0.10 $W/m^2$) to NCB (0.013 $W/m^2$) with that of NCNB (0.077 $W/m^2$) to NCB shows that the global effect of convective scavenging efficiency decrease is larger than photo-bleaching. A similar conclusion can be obtained by comparing the global mean DRE relative change ICNB (0.26 $W/m^2$) to NCNB with that of ICNB to ICB. The DRE relative

change from ICB to ICBB (0.030 $W/m^2$) indicates the photo-bleaching effect of convectively transported BrC is larger than the enhancement of BrC convective transport. The 0.013 $W/m^2$ DRE in the NCB simulation is lower than previous model studies considering the photo-bleaching effect (Wang et al., 2018; Brown et al., 2018). In the NCB simulation, remote BrC concentrations are mostly affected by the threshold for photo-bleaching, which is 6% in this study (Forrister et al. 2015) in comparison to 25% in Wang et al. (2018) and Brown et al. (2018), causing the difference in the global DRE estimates with

photo-bleaching between this work and previous studies. The 0.077 $W/m^2$ DRE in the NCNB simulation is comparable to previous studies (Feng et al., 2013; Jo et al., 2016; Wang et al., 2014).

Only ICB simulation results are discussed hereafter. Figure 12 shows the seasonally BrC DRE distributions. The seasonal variation is due primarily to biomass burning. Figure 2 shows that the largest fire emissions in August, September, July, and

June. While fire emissions are mostly in the tropics in SON, burning at northern mid and high latitudes much more pronounced in JJA in addition to tropical burning.

**5.4 Global effects of BrC absorption on the atmosphere**

As found by Zhang et al. (2017), the importance of radiative heating by BrC relative to BC increases with altitude due to convective transport. We compared the difference of average vertical profiles of BrC to BC heating rate ratio in Figure 14. Over regions directly affected by convection, where the convective mass flux is >$10^{-5}$ kg/m$^2$/s, the simulated result is in agreement with Zhang et al. (2017). Globally, the average BrC/BC heating rate ratio is 15% below 500 hPa and is 44% above 500 hPa. In deep convection regions, the average BrC/BC heating rate ratio is 60% below 500 hPa and 118% above 500 hPa, indicating in deep convection regions, atmospheric heating of BrC is stronger than that of BC. In comparison, over the regions not directly affected by deep convection, the globally averaged BrC/BC heating rate ratio increases from 9% at surface level to 53% in the upper troposphere. Geographically, Figure 15 shows that the difference between BrC and BC is particularly large over the tropics.

The heating rate from aerosols, especially its vertical profile, has significant implications on cloud dynamics (e.g., Bond et al., 2013), and can also induce feedback from regional circulation and the planetary boundary layer dynamics (e.g., Ramanathan and Carmichael, 2008). We conducted three 5-year free-running model simulations for the present day (the year 2010). In the control simulation, both BrC and BC heat were included. We then conducted sensitivity simulations in which only BrC or BC heating was included. By comparing the sensitivity simulation results to the control simulation, we diagnosed the climate response to differential BrC and BC heating.

The latitudinal and vertical difference between BrC and BC heating (Figures 14 and 15) implies that BrC heating is more dominant in the tropics and tends to decrease the vertical gradient of temperature. Allen et al. (2012) suggested that light-absorbing aerosols may contribute to tropical expansion. They did not consider the effect of BrC, the atmospheric heating effect of which is more concentrated in the tropics than BC heating (Figure 15). Using the latitude where the Mean Meridional Circulation (MMC) at 500 hPa becomes zero on the poleward side of the subtropical maximum to diagnose the boundary of the tropics (Zhou et al., 2011), we estimated a 1.0° ± 0.9° latitude of tropical expansion due to BrC heating in comparison to a 1.2° ± 2.9° expansion due to BC heating. The large uncertainty comes from 5 years of free-running simulations and the relatively low model spatial resolution. Another effect from BrC heating is the decrease of deep convective mass flux over the upper troposphere (Feingold et al., 2005; Yoshimori and Broccoli, 2008). We estimated a decrease of deep convective mass flux by $4.41 \times 10^{-5}$ kg/m$^2$/s or 4.1% over the tropics, which is about one-third of the corresponding BC heating effect ($1.52 \times 10^{-4}$ kg/m$^2$/s or 12.9%).

Hodnebrog et al. (2016) suggested that biomass burning aerosols suppress precipitation regionally due primarily to aerosol-cloud interactions. On a global basis, BrC heating reduces precipitation by 0.9% ± 7.0%, which is about 60% of the precipitation reduction by BC simulated in the model. Over the tropical region with high-intensity convection and precipitation, BrC heating decreased precipitation by 3.9% ± 17.8%, similarly to that from BC heating (4.0% ± 17.1%). The effect of BrC heating on tropical precipitation (~100%) is larger than on convective mass flux (~1/3) relative to BC heating because of the stronger BrC than BC heating in the upper than lower troposphere (Figures 14 and 15). BrC heating decreased precipitation by 0.3% ± 10.7% in the northern mid and high latitudes, which is much lower than the effect of BC heating (-4.8 ± 13.5%).

## 6 Conclusions

Light-absorbing aerosols, including BC and BrC, have significant impacts on the global radiative balance. Observational evidence emerged from the DC3 mission of large enhancements of BrC relative to BC over biomass burning regions (Zhang et al., 2017). We developed a module for simulating the effects of brown carbon light absorption in CESM CAM5 and conducted two sets of model experiments, 2010 with nudged meteorological fields and 5-year free-running simulations. Compared to previous studies which did not consider differential convective transport of BrC and BC, the simulated BrC DREs without (NCNB) and with (NCB) photo-bleaching are comparable to previous studies (Feng et al., 2013; Jo et al., 2016; Wang et al., 2018; Brown et al., 2018). However, evaluations with DC3, SEAC[4]RS, and HIPPO observations suggested that the model could simulate the observed concentrations of BrC and BC, although model biases were also found. Reducing the convective scavenging efficiency and including photo-bleaching were necessary to simulate the observed BrC distributions. Globally, the former effect is larger than the latter on simulated BrC absorption. Since the two factors have opposite effects on simulated BrC DRE, our best estimation of global DRE of BrC is 0.10 W/m$^2$.

The BrC DRE is estimated to be 25% of that of BC. Since biomass burning emissions tend to occur during the warm seasons when solar insolation is strong and these emissions tend to occur in the tropics when convective transport is active, the proportional contribution to BrC DRE by biomass burning emission is larger than its fraction in the total emissions. For example, biofuel BrC emissions are seasonal and occur in mid and high latitudes, the combination of BrC absorption and solar radiation of biofuel BrC is less than biomass burning BrC for a unit of BrC emission. Relative to BC DRE, BrC DRE tends to be larger in the tropics due to different emission distributions and larger BrC levels in the upper troposphere. BrC heating reduces global precipitation by 0.9%, about 60% of the BC induced precipitation decrease. Over the tropics, the reduction of precipitation due to BrC heating is similar to BC heating, but its effect on reducing tropical convective mass flux is only ~1/3 of BC heating because BrC heating is strongly skewed to high altitudes compared to BC heating. Consequently, the effect of BrC heating on tropical expansion is comparable to BC heating.

There are still considerable uncertainties in modeling BrC absorption and its effects in the atmosphere. Parameterizations of emissions, photo-bleaching, and convective transport of BrC all require more field and laboratory observations. The uncertainty of model simulated BC also affects the comparison between the DRE and heating of BC and BrC. For example, the model overestimates of BC in the middle and upper tropical troposphere (Figure 5) may lead to an underestimate of the BrC to BC DRE ratio over the remote tropics. The modeling result of stronger atmospheric heating by BrC than BC over the tropical free troposphere in this study are also subject to these uncertainties. Field measurements over tropical convective regions during periods of biomass burning are critically needed to further improve our understanding of BrC processes and its climate effects. Continuous model development by coupling BrC related processes and climate effects into an interactive climate-fire-ecosystem model (Zou et al., 2019) in CESM would also benefit future projections of climate radiative forcing given large impacts of fire emitted BrC in the tropics found by this study.

*Data availability.* NASA DC3 and SEAC[4]RS missions and are available to the general public through the NASA data archive (https://www-air.larc.nasa.gov/cgi-bin/ArcView/dc3 and https://www-air.larc.nasa.gov/cgi-bin/ArcView/seac4rs). MODIS Terra level 3 monthly dataset (MOD08_M3) is available via NASA LAADS Archive (https://ladsweb.modaps.eosdis.nasa.gov/archive/allData/61/MOD08_M3/). HIPPO merged 10-second data is available via CDIAC HIPPO data archive (https://hippo.ornl.gov/data_access). AERONET AOD measurements are available in https://aeronet.gsfc.nasa.gov/new_web/download_all_v3_aod.html. The CAM5 model results are available from the corresponding author upon request.

*Competing interests.* The authors declare that they have no conflict of interest.

*Author contribution.* Yuzhong Zhang and Rodney J. Weber provided the original idea. Aoxing Zhang, Yuhang Wang and Yufei Zou designed the model experiments. Yongjia Song and Ziming Ke conducted and provided the model input data. Rodney J. Weber and Yuzhong Zhang conducted and analyzed aircraft measurement data. Aoxing Zhang carried out the model experiments and prepared the manuscript with contributions from all co-authors.

*Acknowledgments.* This work was supported by the National Science Foundation (NSF) through grant 1243220. We would like to acknowledge high-performance computing support from Cheyenne (Computational and Information Systems Laboratory, 2017) and Yellowstone (ark:/85065/d7w3xhc) provided by National Center for Atmospheric Research (NCAR)'s Computational & Information Systems Lab (CISL), sponsored by the National Science Foundation. RJW was supported through the ATom project by the National Aeronautics and Space Administration under grant NNX15AT90G. We thank Haviland Forrister for helpful discussion.

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

**Table 1. BrC emission sources (TgC yr$^{-1}$) of this and previous studies.**

|  | This work | Jo et. al. (2016) | Wang et al. (2014) |
|---|---|---|---|
| Primary biomass burning source | 3.6 | 3.0±1.7 | 8 for primary sources |
| Primary biofuel source | 3.1 | 3.0±1.3 | |
| Secondary formation | 4.1 | 5.7 | 3.2 |

**Table 2. BrC sensitivity simulations**

| CAM run name | NCNB (base model) | ICNB | NCB | ICBB | ICB (best model) |
|---|---|---|---|---|---|
| Reduced BrC convective wet scavenging | No | Yes | No | Yes | Yes |
| Photo-bleaching of convective transported BrC | No | No | Yes | Yes | No |
| Photo-bleaching of non- convective transported BrC | No | No | Yes | Yes | Yes |

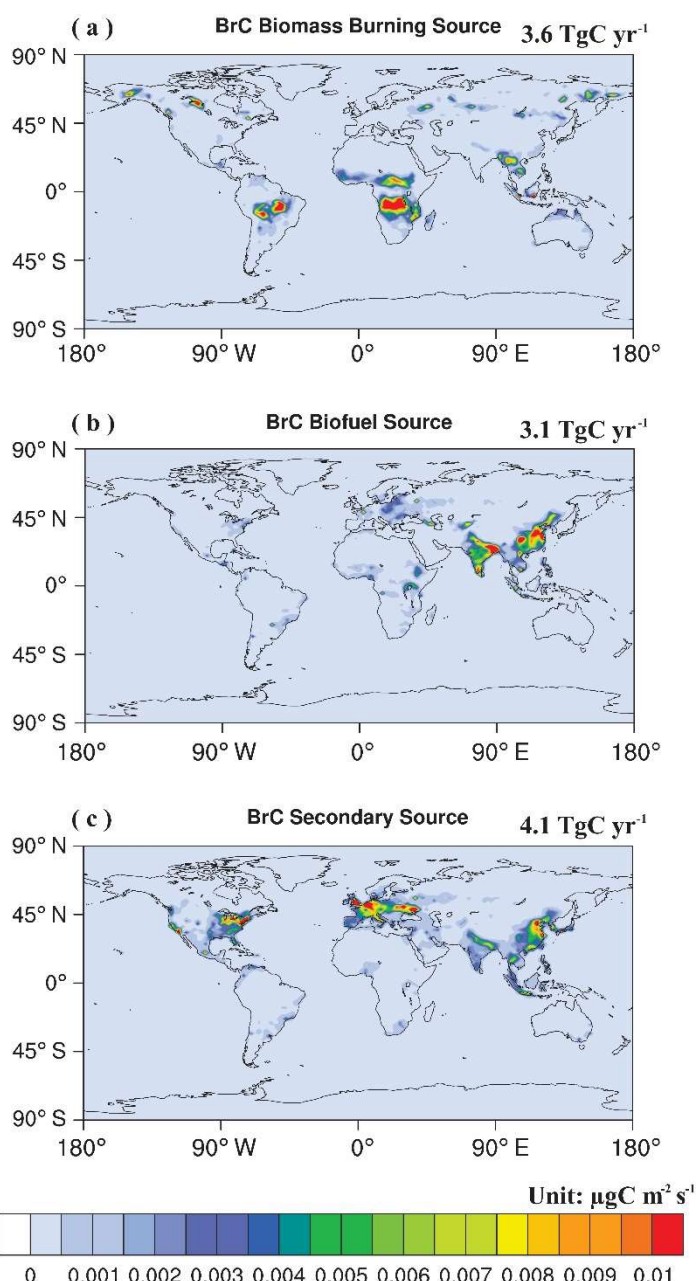

**Figure 1.** Spatial distributions of global emissions of BrC from biomass burning (a), anthropogenic biofuel combustion (b) and secondary formation (c) in 2010. Unit is μg C m$^{-2}$ s$^{-1}$. The total emission is 3.6, 3.1, and 4.1 Tg C yr$^{-1}$ for biomass burning, biofuel, and secondary formation, respectively.

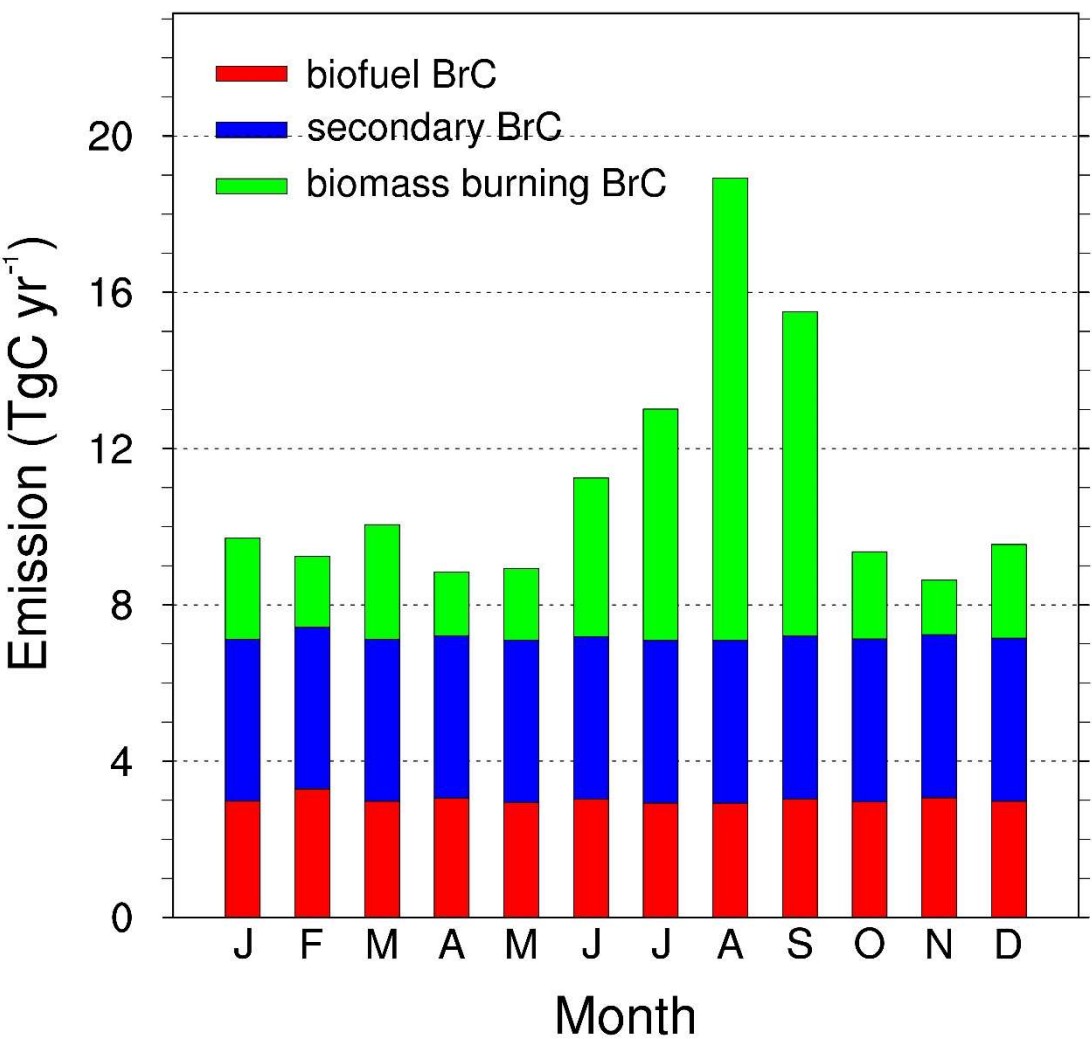

**Figure 2**. Monthly mean global BrC emission rates (Tg C yr⁻¹) in 2010. Green, blue, and red bars represent the emissions from biomass burning, biofuel combustion, and secondary BrC formation, respectively.

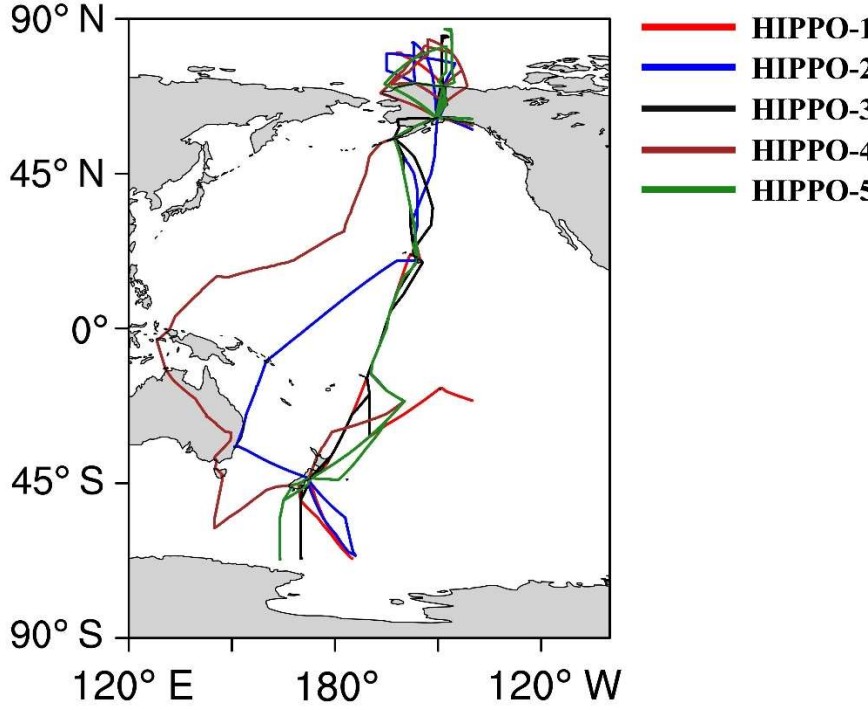

**Figure 3**. Flight track of the 5 HIPPO missions. Colored lines in red, blue, black, brown, green represent flight track of HIPPO-1 to HIPPO-5 respectively. Flights over continental North America east of 140°W were not included in this study.

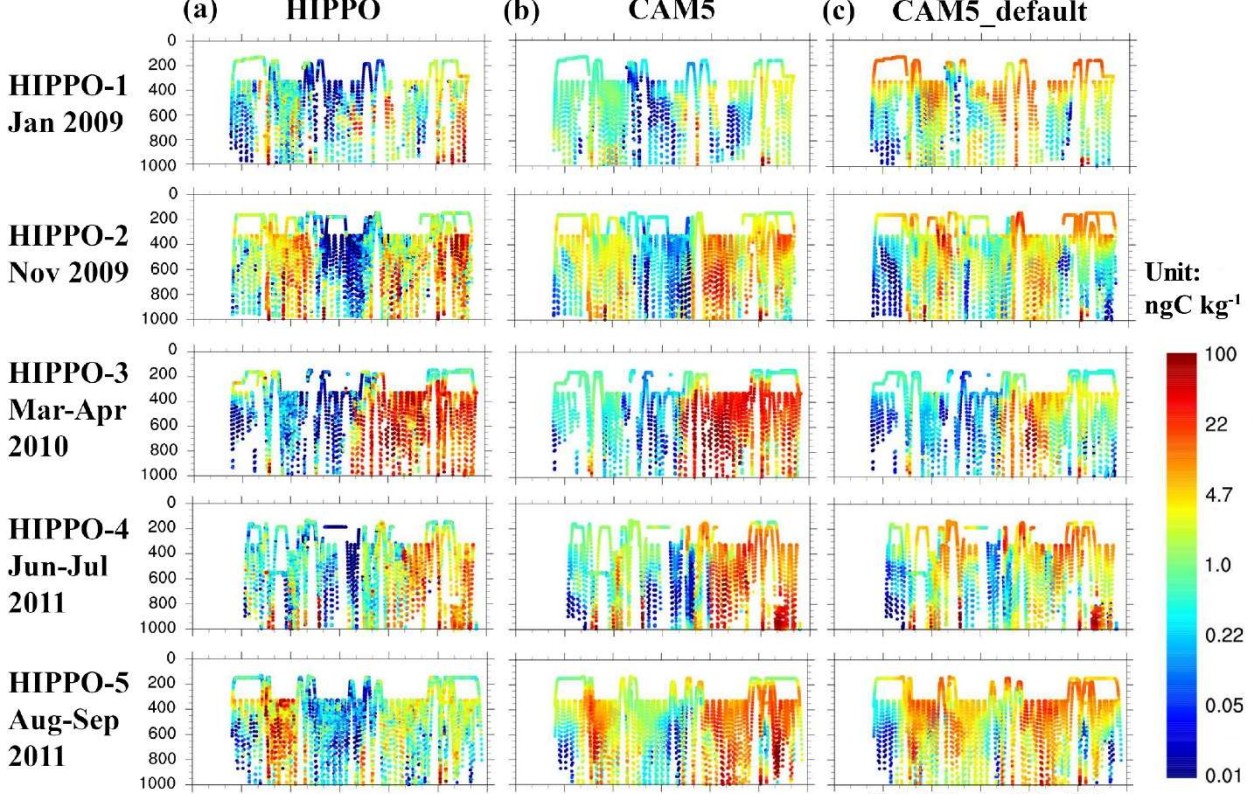

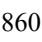

**Figure 4.** Comparison of HIPPO BC (ng C kg⁻¹) measurements (a), simulated BC data from the modified CAM5 model (b) and simulated BC data from the default CAM5 model (c) during HIPPO mission 1-5. The 5 rows from top to bottom are HIPPO-1 (Jan 2009), HIPPO-2 (Nov, 2009), HIPPO-3 (Mar-Apr, 2010), HIPPO-4 (Jun-Jul, 2011), and HIPPO-5 (Aug-Sep, 2011), respectively. Measurement data along the flight tracks of Fig. 3 are 1-min averages. Model data are selected corresponding to the location and time of aircraft measurements.

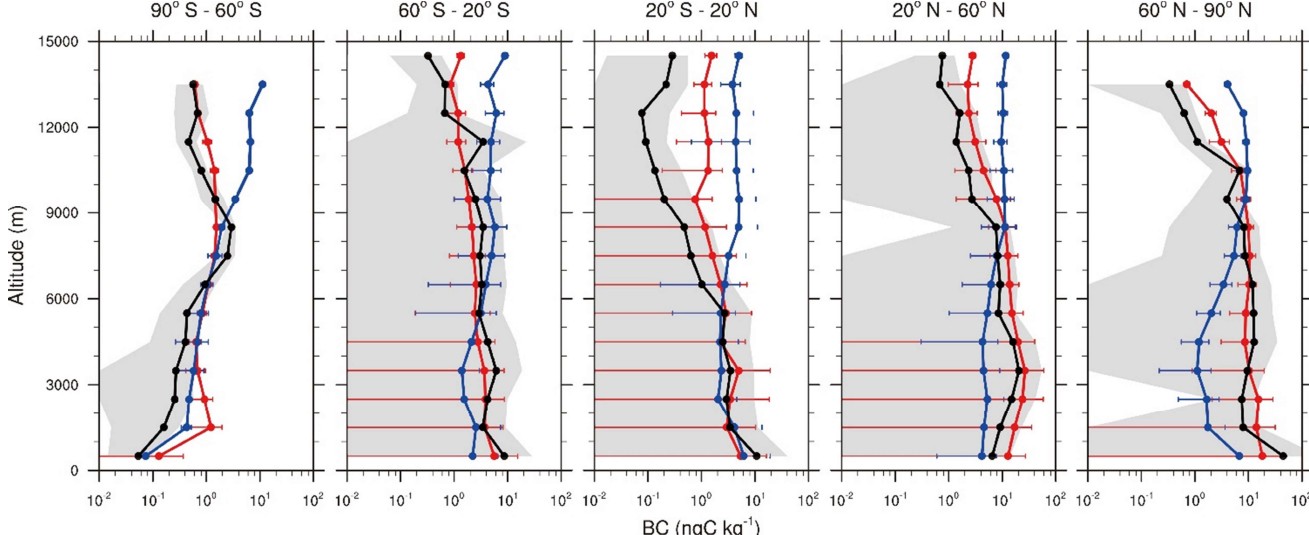

**Figure 5**. Comparison of observed and simulated BC vertical profiles during HIPPO missions for the latitude bins of 90° S-60° S, 60° S-20°S, 20° S-20° N, 20° N-60°N and 60° N-90°N. Black lines and shaded areas show the means and standard deviations of the observations binned in 1-km intervals, respectively. The colored vertical lines and horizontal bars show the means and standard deviations of the default (blue) and modified CAM5 results (red), respectively.

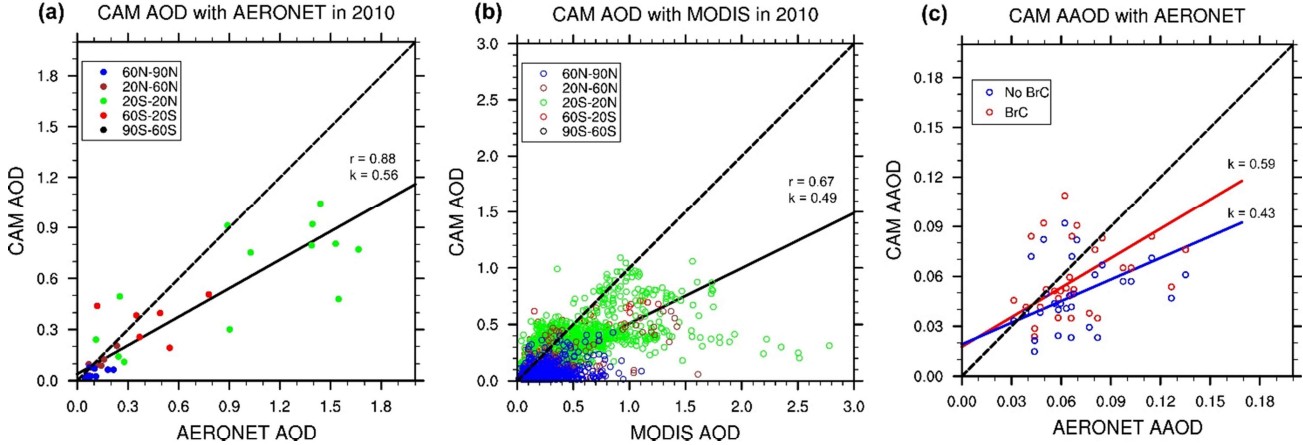

**Figure 6**. Comparison of monthly mean AOD and AAOD data in 550 nm for fire dominated months and regions (Figure S1(a)) of model simulations with the observations for (a) 2010 AERONET AOD; (b) 2010 MODIS AOD and (c) 2005-2014 AERONET AAOD. For (a) and (b), model data correspond to the same time and location of the observations. The data points in (a) and (b) are color-coded as a function of latitude. The solid line denotes a PC regression line and the dashed line

denotes the 1:1 reference line. For (c), monthly mean values of model data corresponding to AERONET AAOD observations are shown. The solid lines denote PC regression lines for model results with and without BrC absorption, and the corresponding regression slope (*k*) values are shown. The dashed line denotes the 1:1 reference line.

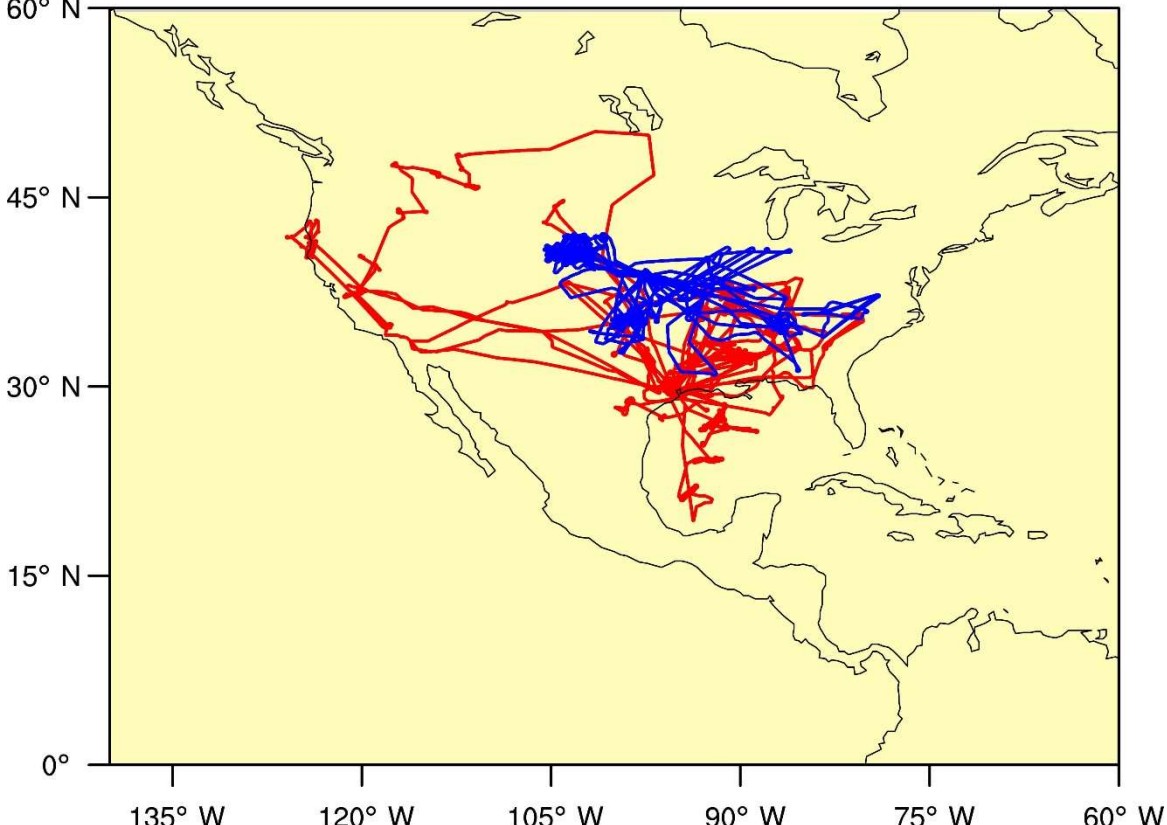

**Figure 7**. Flight tracks of SEAC[4]RS (red) and DC3 (blue) field experiments.

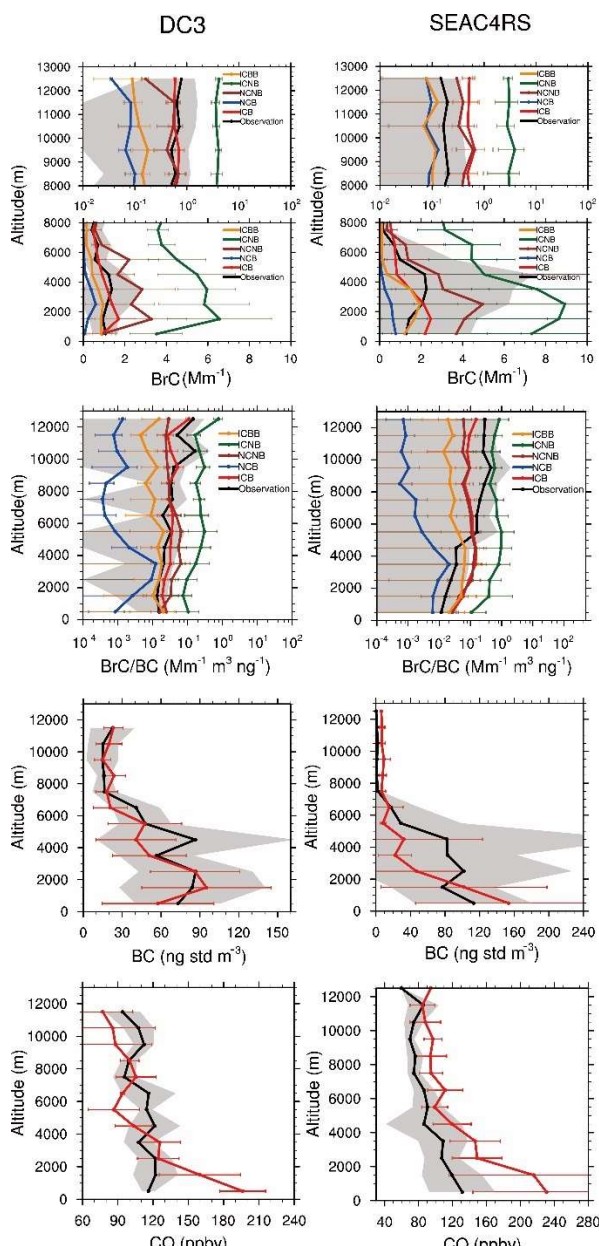

**Figure 8**. Comparison between observed and simulated vertical profiles of BrC absorption at 365 nm, the ratio between BrC absorption at 365 nm and BC (BrC/BC), and concentrations of BC and CO for the DC3 (left column) and SEAC[4]RS (right column) missions. Black lines and shaded areas show the means and standard deviations of the observations binned in 1-km intervals, respectively. The colored vertical lines and horizontal bars show the means and standard deviations of corresponding model results, respectively. Model sensitivity simulations of BrC are listed in Table 2. The difference among simulated BC and CO vertical profiles is negligible and the ICB simulation results are shown.

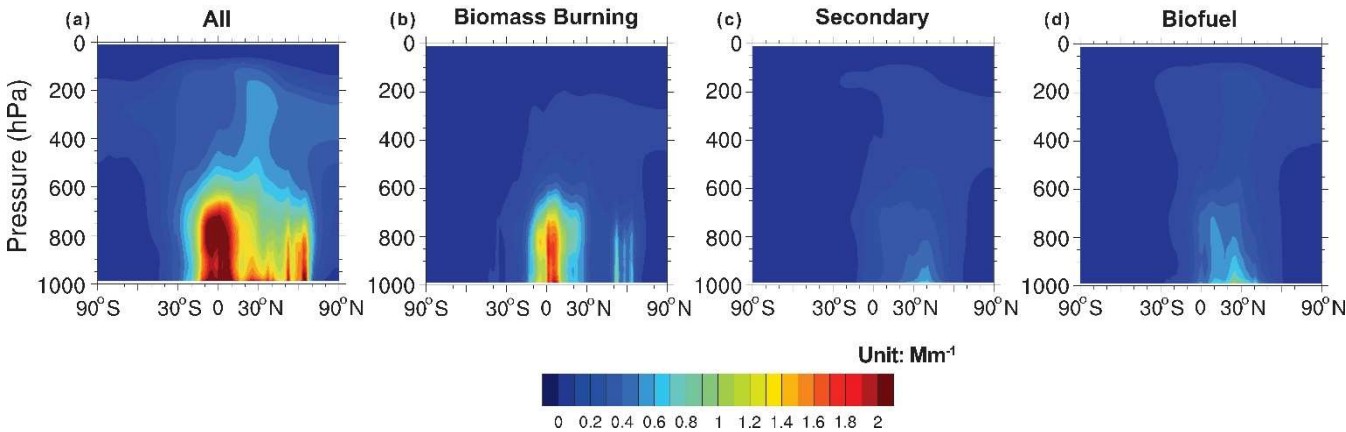

**Figure 9**. Simulated zonal averaged annual mean BrC absorption at 365 nm (Mm$^{-1}$) for (a) all sources, (b) biomass burning emissions, (c) secondary BrC formation, and (d) biofuel BrC emissions. Unit is Mm$^{-1}$. Color bar is in log scale.

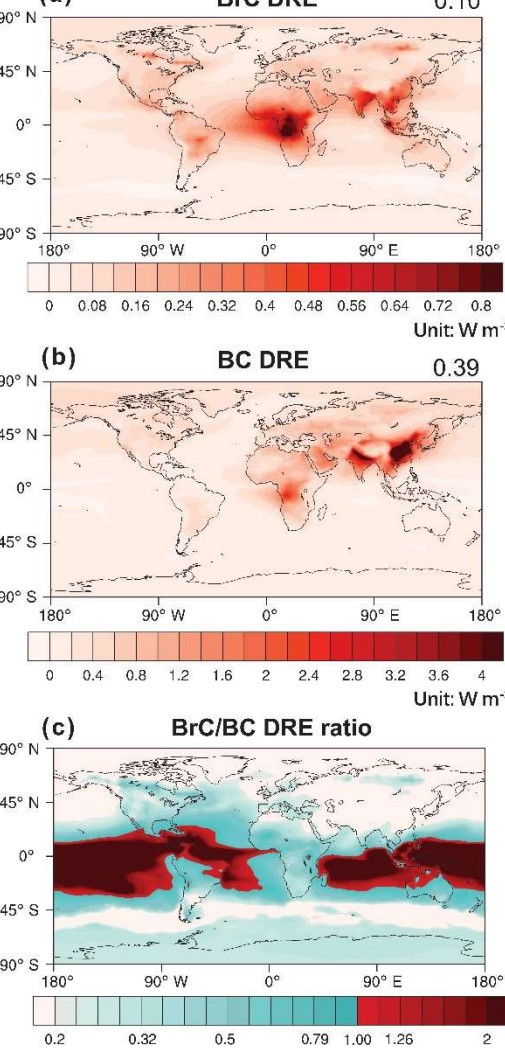

**Figure 10**. Annual averaged global distributions of (a) BC DRE, (b) BrC DRE, and (c) ratio of BrC/BC DRE for 2010. The unit is W m⁻². The global averaged DRE is shown in the upper right corner. In (c), BrC/BC DRE ratios larger than 1.0 are specified by a different color bar.

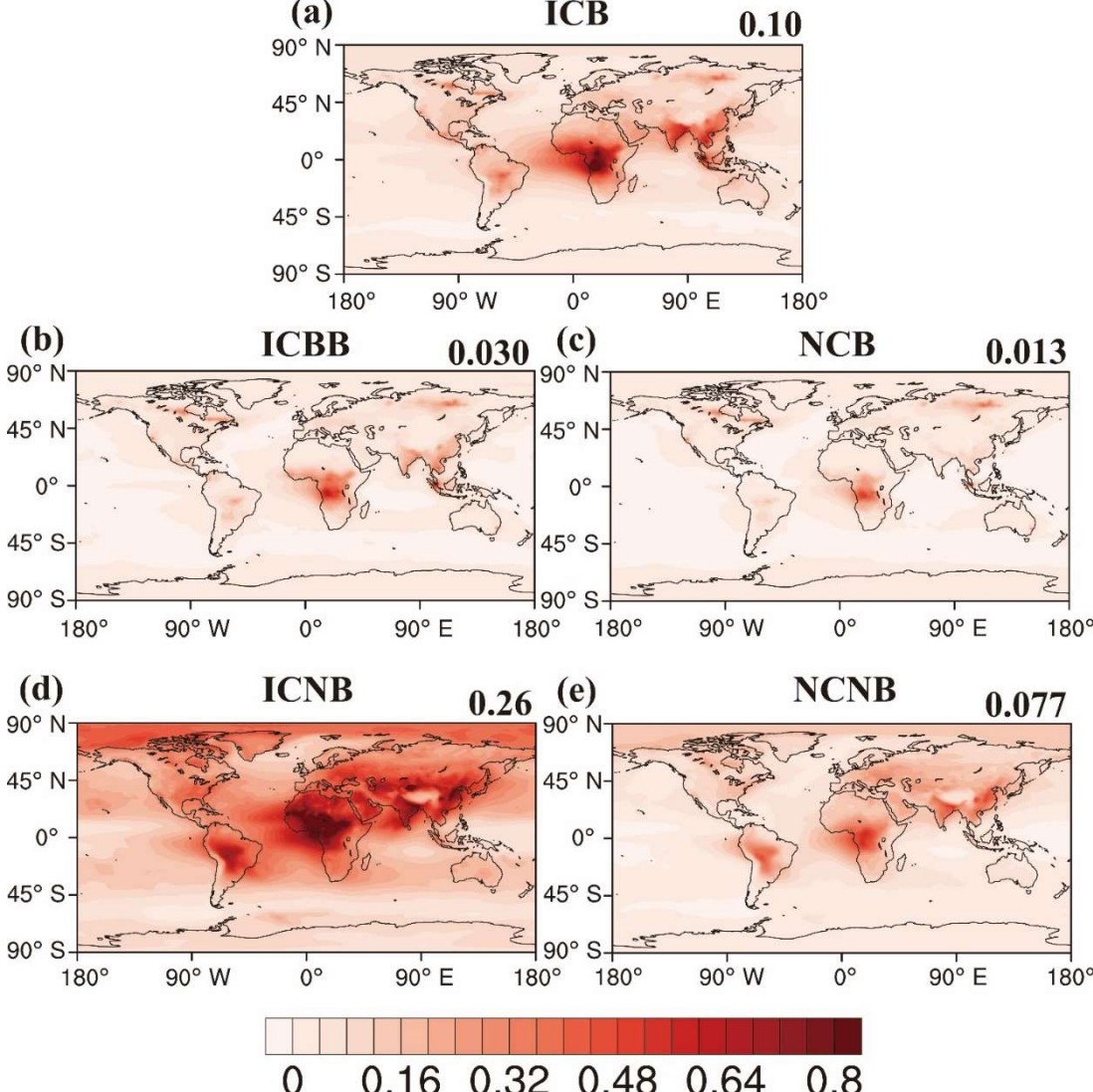

900

**Figure 11**. Annual averaged global distributions of BrC DRE for all sensitivity simulations (Table 2). The unit is W m$^{-2}$. The global averaged DRE is shown in the upper right corner.

905

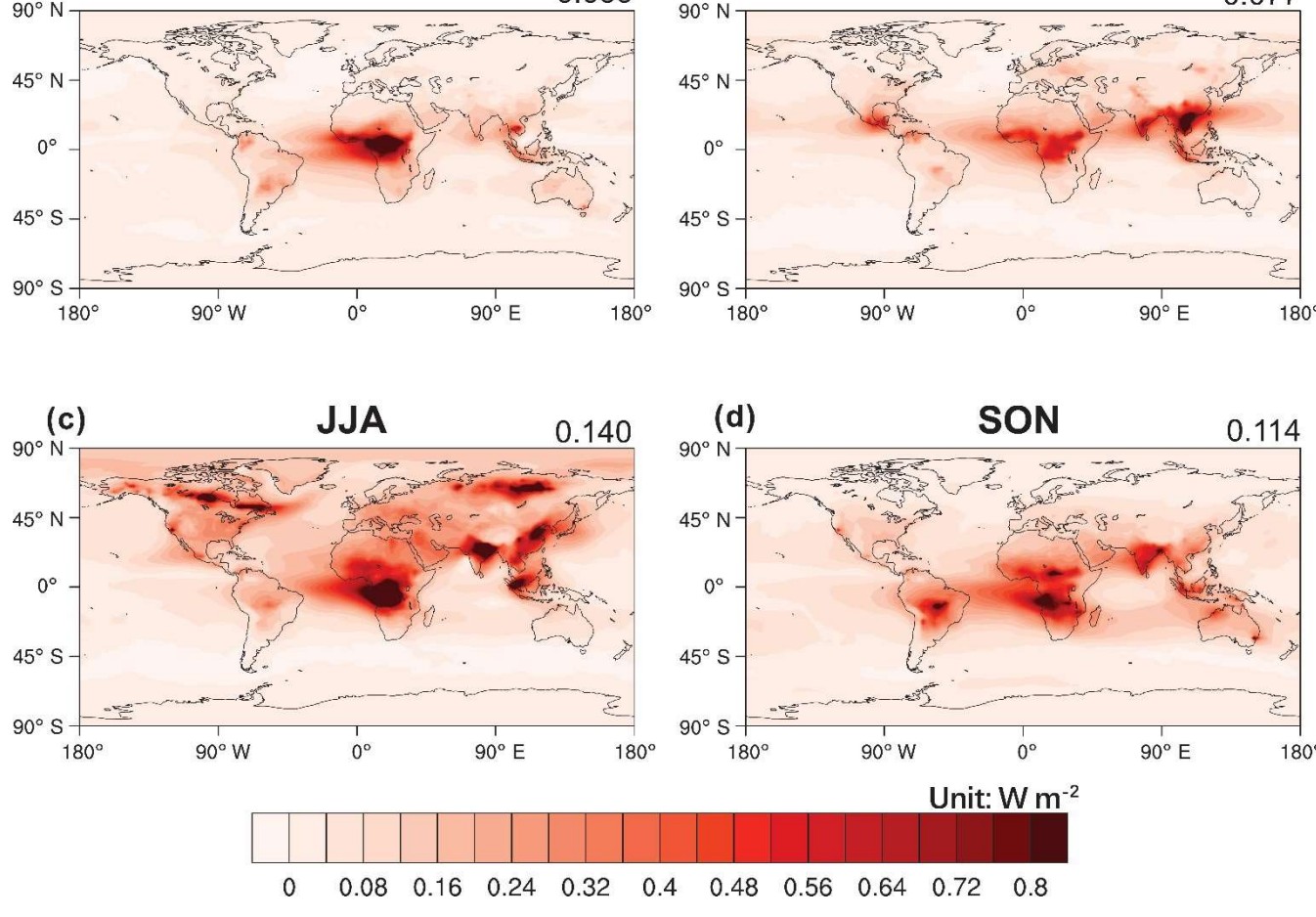

**Figure 12**. Same as Fig. 10 but for seasonal global DRE distributions of BrC for (a) DJF, (b) MAM, (c) JJA, (d) SON in the ICB simulation.

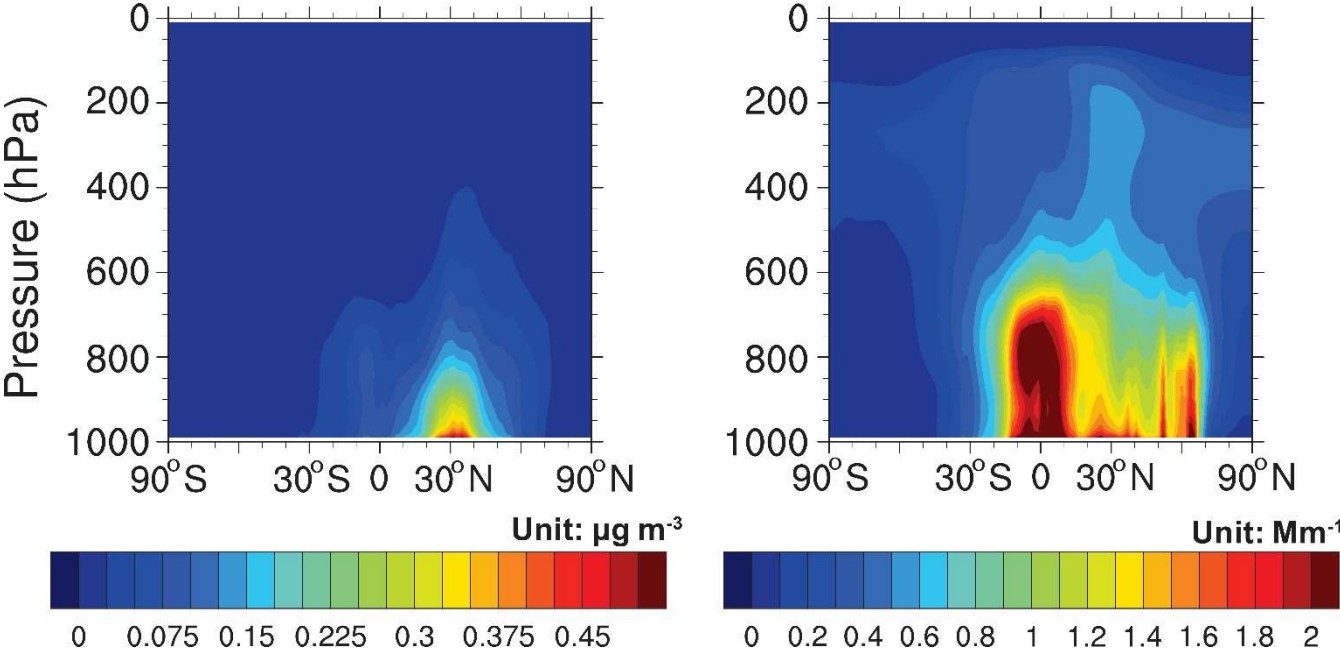

**Figure 13**. Global zonal mean distributions of (a) BC mass concentrations (μg std m$^{-3}$) and (b) BrC absorption at 365 nm (Mm$^{-1}$) for 2010.

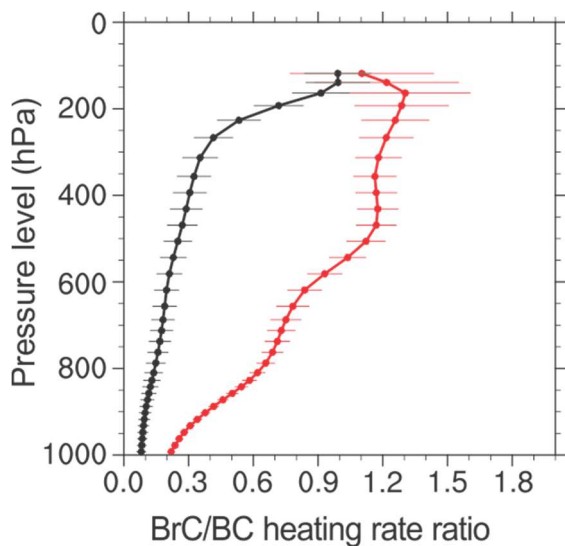

**Figure 14**. Global averaged vertical profile of BrC to BC heating rate ratio for 2010. The black and red lines are the average profiles for regions without and with deep convection events, respectively. Standard deviations are indicated by the horizontal bars.

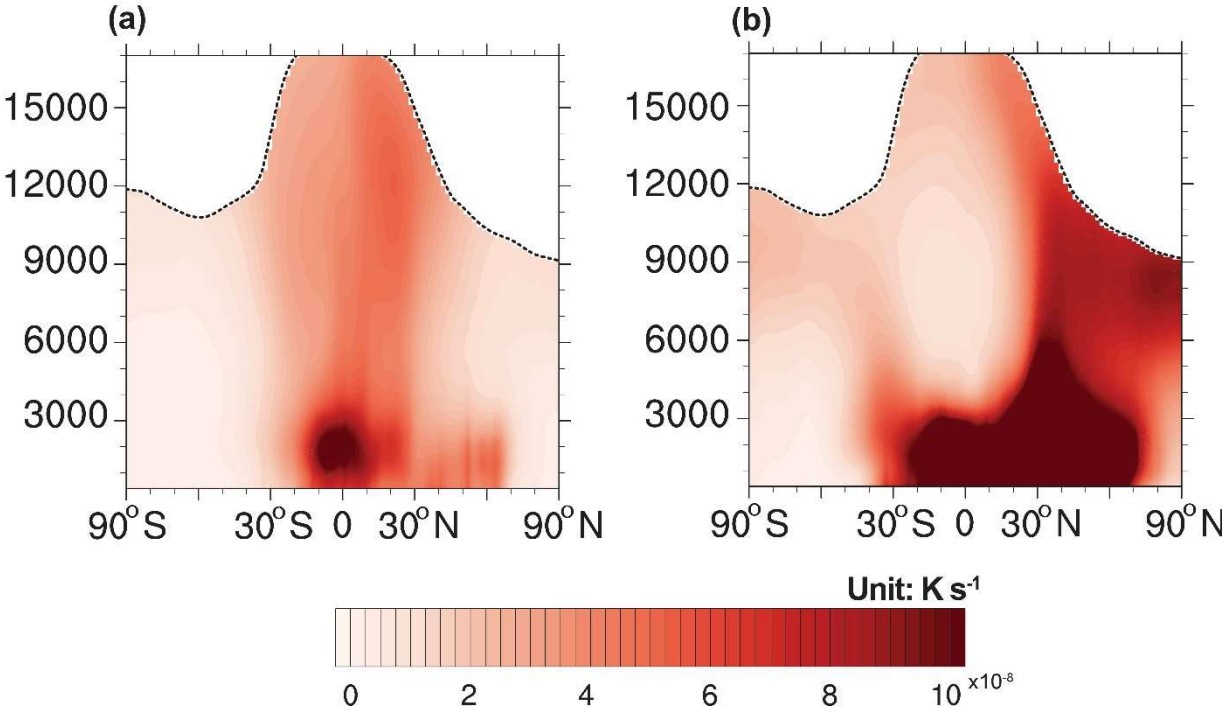

**Figure 15**. Global zonal mean distributions of heating rate of (a) BrC and (b) BC for 5 years of present-day simulations. The dash line denotes the tropopause.