# Peer review of "Modeling global radiative effect of brown carbon: A potentially larger heating source in the tropical free troposphere than black carbon"

_Atmospheric Chemistry and Physics, 2019_

## Short Comment (SC1) · 18 Jul 2019

Thankyou for this interesting and well-written paper. I have a couple of questions/comments that I think would help clarify some details

L125: You talk about the BrC optical properties, but what MAE/optical properties did you use for BC? This would affect the BrC/BC DRE ratio

L317 - 323: Can you comment on the mechanism for BrC being transported further than BC? This seems quite important to your conclusions, if you are saying they are from the same source in the tropics. You say it's a "simulated feature" that BrC DRE is

larger than BC in some regions- do you expect this to be a real feature?

Please can you also add a table explaining what all the different simulations are (ICB, ICNB, CNN etc.)

---

## Author Comment (AC1) · 30 Jul 2019

Dear Dr. Taylor,

Thank you for your helpful questions and comments.

1. The BC MAE values used in this study are 10 and 8.1 m2/g for 345- 442 nm and 442-625 nm, respectively (Knox et al., 2009; Bond and Bergstrom, 2006). We will add this information in the revision.

2. We don't have a solid physical explanation about BrC convective transports, but observations in SEAC4RS and DC-3 clearly show evidence for the process. There are

some mechanisms related to the further transport of BrC. Zhang et al. (2017) showed BrC relative to BC enhancements for smoke plumes undergoing convection and suggested that in-cloud heterogeneous processing may produce BrC. Browne et al. (2019) also suggested that a fraction of BrC produced from heterogeneous oxidation by ozone is free from the bleaching effect and BrC produced from heterogeneous oxidation of OH has a longer lifetime. In addition, hydrophobic OC, such as humic-like substances (HULIS), is more likely to have high light-absorption compared to hydrophilic OC (Hoffer et al., 2006). BrC with high molecular weight dominates the aged biomass burning plume (Wong et al., 2017, Wong et al., 2019). Since higher molecular weight compounds have weaker hygroscopicity (Dinar et al., 2007), and it is harder to activate hydrophobic OC in cloud, less BrC is removed in deep convection.

We simulated that BrC DRE can be larger than BC DRE over strong deep convection regions such as the remote tropics. We expect this to be a real feature, and future measurements over the remote tropics will provide some evidence about the remote upper troposphere BrC.

3. Table. 2 in the manuscript describes the differences among the sensitivity runs (NCNB, ICNB, NCB, ICBB, ICB).

References

Bond, T.C. and Bergstrom, R.W., 2006. Light absorption by carbonaceous particles: An investigative review. Aerosol science and technology, 40(1), pp.27-67.

Browne, E.C., Zhang, X., Franklin, J.P., Ridley, K.J., Kirchstetter, T.W., Wilson, K.R., Cappa, C.D. and Kroll, J.H., 2019. Effect of heterogeneous oxidative aging on light absorption by biomass burning organic aerosol. Aerosol Science and Technology, 53(6), pp.663-674.

Dinar, E., Taraniuk, I., Graber, E.R., Anttila, T., Mentel, T.F. and Rudich, Y., 2007. Hygroscopic growth of atmospheric and model humic‐like substances. Journal of

[Figure]

Geophysical Research: Atmospheres, 112(D5).

Hoffer, A., Gelencsér, A., Guyon, P., Kiss, G., Schmid, O., Frank, G.P., Artaxo, P. and Andreae, M.O., 2006. Optical properties of humic-like substances (HULIS) in biomass-burning aerosols. Atmospheric Chemistry and Physics, 6(11), pp.3563-3570.

Knox, A., Evans, G.J., Brook, J.R., Yao, X., Jeong, C.H., Godri, K.J., Sabaliauskas, K. and Slowik, J.G., 2009. Mass absorption cross-section of ambient black carbon aerosol in relation to chemical age. Aerosol science and technology, 43(6), pp.522-532.

Wong, J.P., Nenes, A. and Weber, R.J., 2017. Changes in light absorptivity of molecular weight separated brown carbon due to photolytic aging. Environmental science & technology, 51(15), pp.8414-8421.

Wong, Jenny PS, Maria Tsagkaraki, Irini Tsiodra, Nikolaos Mihalopoulos, Kalliopi Violaki, Maria Kanakidou, Jean Sciare, Athanasios Nenes, and Rodney J. Weber. "Atmospheric evolution of molecular-weight-separated brown carbon from biomass burning." Atmospheric Chemistry and Physics 19, no. 11 (2019): 7319-7334.

Zhang, Y., Forrister, H., Liu, J., Dibb, J., Anderson, B., Schwarz, J.P., Perring, A.E., Jimenez, J.L., Campuzano-Jost, P., Wang, Y. and Nenes, A., 2017. Top-of-atmosphere radiative forcing affected by brown carbon in the upper troposphere. Nature Geoscience, 10(7), p.486.

---

## Referee Comment (RC1) · Anonymous Referee #1 · 4 Aug 2019

In this study, the authors implement a brown carbon scheme into a global model, evaluate a series of simulations with varying assumptions about photobleaching and convective transport against observations from HIPPO, SEAC4RS and DC3, and then estimate the heating rates and DREs from their simulation which best fits observations. This is certainly an interesting topic which warrants further modeling studies to explore the impacts of large uncertainties in our understanding of the properties and evolution of BrC in the atmosphere. There are however two major issues in this manuscript that should be addressed prior to publication (as well as more minor issues described below):

1. This modeling study shares significant methodological overlap with previous modeling efforts for BrC, particularly Wang et al, ACP, 2018 and Brown et al., ACP, 2018, the first of which compares to the same BrC observations used here and the second of which uses the same model. Both of these previous model studies explore photobleaching. The primary novelty of this study is therefore the focus on convective transport. The authors should therefore be careful not to overstate the novelty of their work, and acknowledge and contrast to the existing literature throughout (Introduction, Results, Conclusions), particularly how assumptions made in this study might differ from these previous studies, why different assumptions might have been implemented, and how this impacts the comparison with these previous studies.

2. While the heating rate conclusions are the most interesting aspect of the manuscript, the results are substantially overstated. Figure 13 shows that BrC heating rates barely exceed those of BC in the UT in convective regions. Given that uncertainties on the simulation (convective parameterization, removal, optical properties, etc, etc, etc) are large, the authors cannot state with high confidence that the heating rate from BrC exceeds BC. In particular, given that this study does not include any observational evaluation in the tropics, where the authors suggest this effect is most important, this conclusion is unsupported. The authors should temper the discussion of these results. Similarly, the manuscript title should be modified to eliminate overstatement of the results.

Additional Comments

1. Line 70: missing name on reference

2. Lines 91-95: specify meteorological years simulated

3. Section 2.2: Should discuss how many different emission factors (i.e. biomes) are used in the inventory and whether their resulting BrC inventory adequately represents the variability in fuels.

4. Section 2.2: Given that the authors rely on comparisons between BrC and BC later in the text, they should include details on BC aging and optical properties in this section.

5. Lines 130 and 133: use the same wavelength for MAEs so that they can be compared

6. Line 203-204: specify that this statement applies to the default model

7. Lines 199-215: discussion of BC removal should also reference and compare to approach of global model study of Q. Wang et al. (JGR, 2014).

8. Line 243: requires citation at the end of the sentence.

9. Lines 245-252: it would be useful to discuss why BrC scavenging isn't treated similarly to BC scavenging in each simulation

10. Figure 7: Why don't the authors compare observed and simulated OA mass and BrC absorption directly?

11. Line 262: "During DC3 experiment, respectively." Is not a sentence

12. Line 263: "Both the observations and model simulations show the increase of BrC/BC ratio at in the upper troposphere"; statement is inaccurate, model simulations do not show increase.

13. Lines 261-284: This discussion is a little confusing. It should be clear from the text that the NCB and ICNB simulations are inconsistent with observations, but also that no model simulation captures the increase in BrC/BC as observed, particularly for SEAC4RS.

14. Line 314: I think the authors mean DRE not "radiative forcing" here to be consistent with their earlier discussion.

---

## Referee Comment (RC2) · Anonymous Referee #2 · 9 Aug 2019

This study examines the global direct radiative effect of brown carbon (BrC), using the Community Atmospheric Model version 5 (CAM5). A biomass burning emission inventory and aerosol module for BrC are developed and implemented into CAM5. The direct radiative effect (DRE) due to BrC absorption is estimated, and the results show that the atmospheric heating by BrC is the most significant in the mid and upper troposphere over the tropics, exceeding the BC heating effect. Sensitivity studies are conducted to demonstrate the relative importance of the BrC photo-bleaching effect and convective wet scavenging on the estimated BrC DREs.

Overall, this study presents interesting findings about the global distributions of BrC

and its radiative effects relative to black carbon (BC). However, there are some major concerns about the modeling approach that need to be addressed before consideration for publication.

First, the parameterization of BrC needs clarification and justification. The primary BrC is included with explicit emission inventories scaled to the organic carbon emissions. This assumes BrC as an individual aerosol type emitted separately from other non-absorbing organic aerosol (OA) compounds. Are there observational evidence to support the assumption? It is not clarified if the BrC emissions are then excluded from the total organic aerosol (OA) emissions or not. The double-counting of BrC may lead to artificial increases in OA aerosol mass and total AOD. Similarly, is secondary BrC included as part of the SOA formation or additional SOA formed from aromatics? The rest OA should be non-absorbing, scattering only, and what optical properties are used for the non-absorbing OA? Furthermore, how does this approach differ essentially from Brown et al. (2018)? Both of them parameterize the absorption due to all the OAs (absorbing + non-absorbing) following Sahel et al. (2014). The definition of BrC radiative effect is also a bit confusing. If BrC are emitted separately as an aerosol type with explicit emissions, then the radiative effect of BrC refers to both scattering and absorption.

Second, the main focus of this study is on the BrC heating effect, but it lacks of evaluation of the modeled aerosol absorption. The ratio of BrC absorption to BC is compared with the DC3 and SEAC4RS measurements, however, limited only to the North America. There is no model evaluation of aerosol absorption over the tropics where the BrC heating effect is suggested to be important. It would be useful to know how the modeled aerosol total and spectrally-dependent absorption compare with observations, i.e., AERONET data, with vs without BrC parameterizations. This would provide observational constraints for the calculated heating effects due to BrC and BC. In addition, Figures 5 shows the comparison of modeled AOD with AERONET and MODIS, but it is unclear if the inclusion of BrC improves the simulated AOD or not?

Third, the model sensitivity studies offer limited insights on physical processes. Parameter tuning of the photo-bleaching effects and convective transport of BrC are largely empirical. It is not very convincing, based on Figure 7, to state that the overestimation of BrC/BC ratios in lower troposphere by NCNB (the base mode) and ICNB reflects on the missing of photo-bleaching effect, because it could be due to underestimate of BC. ICB (the best model) that includes the photo-bleaching also overestimates to a similar extent especially when compared with the SEAC4RS data. It is not clear how good or bad these bleaching effects are represented in the model? Or where/when it improves the modeled absorption by BC+BrC? If not, how to improve based on the sensitivity studies conducted? The enhancement of BrC in convective transport is implemented by reducing the wet scavenging efficiency of BrC to match the transported BrC by convective clouds in Zhang et al. (2016). This ignores the impact of uncertainty in model-simulated convective clouds, and possible aqueous formation of secondary BrC in clouds, attributing the BrC changes solely to the scavenging efficiency without justifying it. At least, other factors should be discussed and probably examined in the sensitivity studies.

Other than the main concerns, a few minor comments are also included:

1. Page 5, line 136: why do you use different reference wavelengths for primary and secondary BrC? 2. Page 5, line 137: AAE is highly variable and the estimated BrC absorption is sensitive to AAE. Uncertainty associated with AAE should be discussed and probably examined in sensitivity studies. 3. Page 5, line 139: this equation is confusing. What is the reference? Where is $k\_OA,\lambda$ from the second step to the third? 4. Page 6, line 165: are you using $k\_OA$ from Sahel et al., 2014 for OA, after including explicit BrC emissions? 5. Page 10, section 5.3, equations (6) and (7): I think that the parameterizations in Sahel et al., (2014) are for absorption spectral dependence of the total OA including both absorbing and non-absorbing components. But here they are applied to the absorbing components only? 6. Page 10, line 311: please clarify what two sets of radiative fluxes are used to calculate the BrC DRE

---

## Author Comment (AC2) · 20 Sep 2019

**Referee #1**

Thank you for your helpful comments. We revised the manuscript based on your comments and suggestions.

In this study, the authors implement a brown carbon scheme into a global model, evaluate a series of simulations with varying assumptions about photobleaching and convective transport against observations from HIPPO, SEAC4RS and DC3, and then estimate the heating rates and DREs from their simulation which best fits observations. This is certainly an interesting topic which warrants further modeling studies to explore the impacts of large uncertainties in our understanding of the properties and evolution of BrC in the atmosphere. There are however two major issues in this manuscript that should be addressed prior to publication (as well as more minor issues described below):

        **Author's response:** Thank you for your suggestions. The point-by-point responses to your questions and comments are presented below.

1. This modeling study shares significant methodological overlap with previous modeling efforts for BrC, particularly Wang et al, ACP, 2018 and Brown et al., ACP, 2018, the first of which compares to the same BrC observations used here and the second of which uses the same model. Both of these previous model studies explore photobleaching. The primary novelty of this study is therefore the focus on convective transport. The authors should therefore be careful not to overstate the novelty of their work, and acknowledge and contrast to the existing literature throughout (Introduction, Results, Conclusions), particularly how assumptions made in this study might differ from these previous studies, why different assumptions might have been implemented, and how this impacts the comparison with these previous studies.

**Author's response:** In the revised manuscript, we clarified the differences of our simulations of BrC from previous studies. The implementation of BrC photo-bleaching in Wang et al (2018) and Brown et al (2018) specified a uniform 1-day e-folding time for BrC, and BrC bleaching rate, depending on OH concentrations, until 25% of the original BrC absorption is left. Our approach to BrC photobleaching considers the bleaching effects from different sources. We specify a decay half-life of 12 hours when light is present for primary biomass and biofuel BrC in the daytime until 6% is left and no further photobleaching occurs (Forrister et al., 2015). The half-life for secondary aromatic BrC is specified at 12 hours in daytime (Liu et al., 2016). The revised text is added in Lines 164-177: "Previous modeling of the BrC photo-bleaching effect by Wang et al. (2018) and Brown et al. (2018) applied a 1-day e-folding time for BrC before reaching a threshold of 25% of the original BrC absorption. Our approach to BrC photo-bleaching considers different bleaching effects depending on BrC source. We specify a decay half-life of 12 hours when light is present for primary biomass and biofuel BrC in the daytime until 6% is left and no further photo-bleaching occurs (Forrister et al., 2015) due to stable high molecular weight chromophores (Di Lorenzo and Young, 2015; Di Lorenzo et al., 2017; Wong et al., 2017; Wong et al, 2019). Different components of SOA have different photo-bleaching lifetimes. Aromatic SOA has a half-life of 12-24 hours (Liu et al., 2016; Lee et al., 2014; Zhong and Jang, 2011), limonene SOA has a half-life of <0.5 hours (Lee et al., 2014). Methylglyoxal SOA has a half-life of 90 minutes (Zhao et al., 2015; Wong et al., 2017). Therefore, the half-life for secondary aromatic BrC is specified at 12 hours in daytime until it is completely removed (Liu et al., 2016)."

        In Lines 389-393, we added: "The 0.013 W/m$^2$ DRE in the NCB simulation is lower than previous model studies considering the photo-bleaching effect (Wang et al., 2018; Brown et al., 2018). In the NCB simulation, remote BrC concentrations are mostly affected by the lower threshold for photo-bleaching, which is 6% in this study (Forrister et al. 2015) in comparison to 25% in Wang et al. (2018)

and Brown et al. (2018), causing the difference in the global DRE estimates with photo-bleaching between this work and previous studies.

In the introduction section, Lines 82-83, we added "Model simulation results without considering the differential convective transport and BC and BrC are compared to previous studies."

In the conclusions section, Lines 441-443, we added "Compared to previous studies which did not consider differential convective transport of BrC and BC, the simulated BrC DREs without (NCNB) and with (NCB) photo-bleaching are comparable to previous studies (Feng et al., 2013; Jo et al., 2016; Wang et al., 2018; Brown et al., 2018)."

2. While the heating rate conclusions are the most interesting aspect of the manuscript, the results are substantially overstated. Figure 13 shows that BrC heating rates barely exceed those of BC in the UT in convective regions. Given that uncertainties on the simulation (convective parameterization, removal, optical properties, etc, etc, etc) are large, the authors cannot state with high confidence that the heating rate from BrC exceeds BC. In particular, given that this study does not include any observational evaluation in the tropics, where the authors suggest this effect is most important, this conclusion is unsupported. The authors should temper the discussion of these results. Similarly, the manuscript title should be modified to eliminate overstatement of the results.

**Author's response:** We agree with the reviewer that the uncertainties of BrC module may lead to biases in the simulated BrC heating over the upper troposphere. We updated the title to "*Modeling global radiative effect of brown carbon: A potentially larger heating source in the tropical free troposphere than black carbon*" and add more discussion in Conclusion section admitting the uncertainties of the BrC parameterization.

In the conclusion section, Lines 461-464, we emphasized the uncertainties: "There are still considerable uncertainties in modeling BrC absorption and its effects in the atmosphere. Parameterizations of emissions, photo-bleaching, and convective transport of BrC all require more field and laboratory observations. The modeling result of stronger atmospheric heating by BrC than BC over the tropical free troposphere in this study are subject to these uncertainties. Field measurements over tropical convective regions during periods of biomass burning are critically needed to further improve our understanding of BrC processes and its climate effects…"

**Additional comments**

1. Line 70: missing name on reference
**Author's response:** Thank you for pointing out. Reference added on line 70 (now line 73).

2. Lines 91-95: specify meteorological years simulated
**Author's response:** Our free-running simulations are based on the climatology of 2010. We updated this in line 101.

3. Section 2.2: Should discuss how many different emission factors (i.e. biomes) are used in the inventory and whether their resulting BrC inventory adequately represents the variability in fuels.
**Author's response:** We updated the emission factor variability in the page 4, lines 117-119 "The

different emission factors for tropical forest, temperate forest, boreal forest, savanna, agriculture waste and peat burning are based on Akagi et al. (2011)." We added "The variability of BrC emission rate among biomes therefore depends on the BC to OA emission ratios in the GFED emission inventory." in page 7, lines 208-209.

4. Section 2.2: Given that the authors rely on comparisons between BrC and BC later in the text, they should include details on BC aging and optical properties in this section.
 **Author's response:** We added the information at the end of section 2, lines 133-135.

5. Lines 130 and 133: use the same wavelength for MAEs so that they can be compared
 **Author's response:** We updated MAEs in the same wavelength in line 144. The MAE value at 550 nm for secondary BrC is 0.19 m$^2$/g in the model.

6. Line 203-204: specify that this statement applies to the default model
 **Author's response:** We updated that this statement applies to the default model and all sensitivity runs at lines 241-242.

7. Lines 199-215: discussion of BC removal should also reference and compare to approach of global model study of Q. Wang et al. (JGR, 2014).
 **Author's response:** Wang et al. (2014) updated the model wet scavenging by scavenging hydrophobic aerosols in convective updrafts and scavenging hydrophilic aerosols from cold clouds. We increased the interstitial BC scavenging by a factor of 5 to increase wet scavenging and reduced stratiform liquid-containing cloud based on model evaluations using HIPPO data. We updated this in Page 8, Lines 240-244.

8. Line 243: requires citation at the end of the sentence.
 **Author's response:** We moved the citation to the end of the sentence.

9. Lines 245-252: it would be useful to discuss why BrC scavenging isn't treated similarly to BC scavenging in each simulation
 **Author's response:** We updated the wet scavenging efficiency of BrC based on the convection outflow/inflow ratio discussed in Zhang et al. (2017).

10. Figure 7: Why don't the authors compare observed and simulated OA mass and BrC absorption directly?
 **Author's response:** We updated the comparison of BrC between the model results and observations in the supplement. The change of BrC/BC indicates the different physical chemical properties between BrC and BC, so the ratio of BrC to BC is an important factor when estimating the physical chemical properties of BrC. In figure 7, we compared BrC/BC between the model and the observation.

11. Line 262: "During DC3 experiment, respectively." Is not a sentence
 **Author's response:** We changed the typo in the updated manuscript.

12. Line 263: "Both the observations and model simulations show the increase of BrC/BC ratio at in the upper troposphere"; statement is inaccurate, model simulations do not show increase.
 **Author's response:** The sentence is removed.

13. Lines 261-284: This discussion is a little confusing. It should be clear from the text that the NCB and ICNB simulations are inconsistent with observations, but also that no model simulation captures the

increase in BrC/BC as observed, particularly for SEAC4RS.

**Author's response:** In section 5.1, lines 319-321, we showed in the text that NCB simulation underestimated the BrC/BC ratio in both DC-3 and SEAC[4]RS, and ICNB overestimated the BrC/BC ratio in the observations. In lines 325-326, we now acknowledged that no model simulation captured the increase in BrC/BC as observed in SEAC4RS.

14. Line 314: I think the authors mean DRE not "radiative forcing" here to be consistent with their earlier discussion.

**Author's response:** Corrected. Thank you.

Brown, H., Liu, X., Feng, Y., Jiang, Y., Wu, M., Lu, Z., Wu, C., Murphy, S. and Pokhrel, R., 2018. Radiative effect and climate impacts of brown carbon with the Community Atmosphere Model (CAM5). Atmospheric Chemistry and Physics, 18, 17745-17768, https://doi.org/10.5194/acp-18-17745-2018, 2018.

Forrister, H., Liu, J., Scheuer, E., Dibb, J., Ziemba, L., Thornhill, K. L., Anderson, B., Diskin, G., Perring, A. E., and Schwarz, J. P.: Evolution of brown carbon in wildfire plumes, Geophysical Research Letters, 42, 4623-4630, https://doi.org/10.1002/2015GL063897, 2015.

Liu, J., Lin, P., Laskin, A., Laskin, J., Kathmann, S. M., Wise, M., Caylor, R., Imholt, F., Selimovic, V., and Shilling, J. E.: Optical properties and aging of light-absorbing secondary organic aerosol, Atmospheric Chemistry and Physics, 16, 12815-12827, https://doi.org/10.5194/acp-16-12815-2016, 2016.

Wang, Q., Jacob, D.J., Spackman, J.R., Perring, A.E., Schwarz, J.P., Moteki, N., Marais, E.A., Ge, C., Wang, J. and Barrett, S.R.: Global budget and radiative forcing of black carbon aerosol: Constraints from pole-to-pole (HIPPO) observations across the Pacific. Journal of Geophysical Research: Atmospheres, 119(1), pp.195-206, https://doi.org/10.1002/2013JD020824, 2013.

Wang, X., Heald, C. L., Liu, J., Weber, R. J., Campuzano-Jost, P., Jimenez, J. L., Schwarz, J. P., and Perring, A. E.: Exploring the observational constraints on the simulation of brown carbon, Atmospheric Chemistry and Physics, 18, 635, https://doi.org/10.5194/acp-18-635-2018, 2018.

Zhang, Y., Forrister, H., Liu, J., Dibb, J., Anderson, B., Schwarz, J. P., Perring, A. E., Jimenez, J. L., Campuzano-Jost, P., and Wang, Y.: Top-of-atmosphere radiative forcing affected by brown carbon in the upper troposphere, Nature Geoscience, 10, 486, https://doi.org/10.1038/ngeo2960, 2017.

---

## Author Comment (AC3) · 20 Sep 2019

**Referee #2**

Thank you for your helpful comments. We response to your comments and revise the manuscript based on the comments, described in the following content.

This study examines the global direct radiative effect of brown carbon (BrC), using the Community Atmospheric Model version 5 (CAM5). A biomass burning emission inventory and aerosol module for BrC are developed and implemented into CAM5. The direct radiative effect (DRE) due to BrC absorption is estimated, and the results show that the atmospheric heating by BrC is the most significant in the mid and upper troposphere over the tropics, exceeding the BC heating effect. Sensitivity studies are conducted to demonstrate the relative importance of the BrC photo-bleaching effect and convective wet scavenging on the estimated BrC DREs. Overall, this study presents interesting findings about the global distributions of BrC and its radiative effects relative to black carbon (BC). However, there are some major concerns about the modeling approach that need to be addressed before consideration for publication.

**Author's response:** Thank you for your suggestions. The point-by-point responses to your questions and comments are presented below.

First, the parameterization of BrC needs clarification and justification. The primary BrC is included with explicit emission inventories scaled to the organic carbon emissions. This assumes BrC as an individual aerosol type emitted separately from other non-absorbing organic aerosol (OA) compounds. Are there observational evidence to support the assumption? It is not clarified if the BrC emissions are then excluded from the total organic aerosol (OA) emissions or not. The double-counting of BrC may lead to artificial increases in OA aerosol mass and total AOD. Similarly, is secondary BrC included as part of the SOA formation or additional SOA formed from aromatics? The rest OA should be non-absorbing, scattering only, and what optical properties are used for the non-absorbing OA? Furthermore, how does this approach differ essentially from Brown et al. (2018)? Both of them parameterize the absorption due to all the OAs (absorbing + non-absorbing) following Sahel et al. (2014). The definition of BrC radiative effect is also a bit confusing. If BrC are emitted separately as an aerosol type with explicit emissions, then the radiative effect of BrC refers to both scattering and absorption.

**Author's response:** We understand the reviewer's concerns. While BrC is part of OC in the atmosphere, we can treat them separately using different tracers in model simulations. One can treat BrC as a fraction of OC (as in Brown et al., 2018) but assumptions have to be made (for example, the BrC/OC fraction does not change during transport including convective transport, which should change based on aircraft observations (Zhang et al., 2017)). Our simulation approach is more flexible. There is no double counting problem as long as scattering and absorption are treated separately (through different tracers) in the model. We did not account for the scattering effect of BrC such that absorption of BrC can be compared directly to BC. We added a paragraph at the end of section 3.1, lines 188-192: "One important finding by Zhang et al. (2017) is that wet scavenging of BrC during convection differs from BC and OC. Therefore, BrC is simulated using a different tracer from OC in this work unlike Browne et al. (2018). The BrC property of interest is absorption and we assume that the tracer's optical property is light absorption only (no scattering). Consequently, there is no double counting of OC scattering. In the following analysis, the DRE from BrC is for light absorption only such that it can be directly compared to BC."

Second, the main focus of this study is on the BrC heating effect, but it lacks of evaluation of the modeled aerosol absorption. The ratio of BrC absorption to BC is compared with the DC3 and SEAC4RS measurements, however, limited only to the North America. There is no model evaluation of aerosol absorption over the tropics where the BrC heating effect is suggested to be important. It would be useful to know how the modeled aerosol total and spectrally-dependent absorption compare with observations, i.e., AERONET data, with vs without BrC parameterizations. This would provide observational constraints for the

calculated heating effects due to BrC and BC. In addition, Figures 5 shows the comparison of modeled AOD with AERONET and MODIS, but it is unclear if the inclusion of BrC improves the simulated AOD or not?

**Author's Response:** In general, the effect of aerosol absorption on AERONET AOD is small and this is particularly true for BrC. In Figure 5, we compared AOD from our best model (including BrC absorption) to AERONET and MODIS. The following figure shows modelled AOD without BrC, compared with AERONET and MODIS. It is nearly identical to the results with BrC shown in Figure 5. Quantitively, in fire dominate regions (where BC from fire emissions contributes to >50% of the total BC), BrC contributes ~0.37% of the total AOD and 8.5% of the total absorption aerosol optical depth (AAOD). The increase of AAOD from BrC absorption is more significant than AOD. We added in page 9, lines 274-276, "For these data, the effect of BrC absorption on AOD is small; we estimate that BrC absorption contributes 0.37% of the total AOD and 8.5% of the total absorption aerosol optical depth (AAOD)."

[Figure]

Third, the model sensitivity studies offer limited insights on physical processes. Parameter tuning of the photo-bleaching effects and convective transport of BrC are largely empirical. It is not very convincing, based on Figure 7, to state that the overestimation of BrC/BC ratios in lower troposphere by NCNB (the base mode) and ICNB reflects on the missing of photo-bleaching effect, because it could be due to underestimate of BC. ICB (the best model) that includes the photo-bleaching also overestimates to a similar extent especially when compared with the SEAC4RS data. It is not clear how good or bad these bleaching effects are represented in the model? Or where/when it improves the modeled absorption by BC+BrC? If not, how to improve based on the sensitivity studies conducted? The enhancement of BrC in convective transport is implemented by reducing the wet scavenging efficiency of BrC to match the transported BrC by convective clouds in Zhang et al. (2016). This ignores the impact of uncertainty in model-simulated convective clouds, and possible aqueous formation of secondary BrC in clouds, attributing the BrC changes solely to the scavenging efficiency without justifying it. At least, other factors should be discussed and probably examined in the sensitivity studies.

**Author's Response:** There are uncertainties in the model simulations. In the conclusion section, lines 461-464, we emphasized the uncertainties: "There are still considerable uncertainties in modeling BrC absorption and its effects in the atmosphere. Parameterizations of emissions, photo-bleaching, and convective transport of BrC all require more field and laboratory observations. The modeling result of stronger heating by BrC than BC over the tropical free troposphere in this study are subject to the

uncertainties. The results from this study indicate that field measurements over tropical convective regions during periods of biomass burning are critically needed to further improve our understanding of BrC processes and its climate effects…"

We added the following figure in the supplement (now Figure S2) to show that that BrC in the NCNB model is much higher than the observations in the lower troposphere. Photo-bleaching, which was observed in field observations, is an obvious reason for the overestimated BrC/BC ratio and BrC concentrations in the lower troposphere in the NCNB model.

[Figure]

To further address the reviewer's comments, we added a new paragraph at the end of section 5.1, lines 330-337: "In the ICB simulation, wet scavenging of BrC was reduced relative to BC in order to simulate the observed BrC/BC ratios in DC3 and SEAC[4]RS. The mechanisms are not yet clear due to a lack of laboratory and field observations. Hydrophobic OC, such as humic-like substances (HULIS), is more likely to have high light-absorption compared to hydrophilic OC (Hoffer et al., 2006). BrC with high molecular weight dominates the aged biomass burning plume (Wong et al., 2017, Wong et al., 2019). Since higher molecular weight compounds have lower hygroscopicity (Dinar et al., 2007), and it is harder to activate hydrophobic OC in cloud, less BrC is removed in deep convection. Another possible mechanism is production of BrC through in-cloud heterogeneous processing of fire plumes (Zhang et al., 2017). However, there is no observation data to implement such a mechanism in a model."

Other than the main concerns, a few minor comments are also included:
1. Page 5, line 136: why do you use different reference wavelengths for primary and secondary BrC?
**Author's Response:** We kept the reference wavelengths from the experiments of McMeeking, (2008) and Nakayama et al. (2010) for primary and secondary BrC. To make MAE references agree with each other, we have converted secondary BrC MAE to 550 nm (0.19 m2/g) using the AAE = 5.0 in the updated manuscript section 3.1, line 144.

2. Page 5, line 137: AAE is highly variable and the estimated BrC absorption is sensitive to AAE. Uncertainty associated with AAE should be discussed and probably examined in sensitivity studies.

**Author's Response:** Previous modelling study from Jo et al. (2016) estimated BrC/BC absorption ratio with different AAE of BrC. They found BrC/BC ratio will decrease when BrC AAE increase from 5 to 6.19. We used a BrC AAE of 5.0, which also agrees with the experiment results of Kirchstetter and Thatcher (2012). We discussed the uncertainty of BrC simulation coming from AAE variation in the updated manuscript section 3.1, lines 152-156.

3. Page 5, line 139: this equation is confusing. What is the reference? Where is k_OA, from the second step to the third?
**Author's Response:** The first step from the equation is to convert the imaginary refractive index of OA (BrC + non-absorbing OA) to the imaginary refractive index of BrC. $k_{OA}$ from the second step to the third is based on the Eq.6 in Liu et al., (2013):

$$k = \frac{\rho\lambda \cdot A(\lambda)}{4\pi \cdot c}$$

where k is the imaginary refractive index, $\rho$ is particle density (g m$^{-3}$), $A(\lambda)$ is the light absorption at the wavelength $\lambda$, and c is the mass concentration.

4. Page 6, line 165: are you using k_OA from Sahel et al., 2014 for OA, after including explicit BrC emissions?
**Author's Response:** No. We convert $k_{OA}$ to $k_{BrC}$ using Eq.3, and apply it to the Eq.6 in Liu et al. (2013) to get Eq. 5, and we applied $k_{BrC}$ to the explicit BrC emissions. We added after Eq. (4), lines 200-202, "We computed $k_{OA,550}$ in order to calculate BrC emissions. In the model, the absorption of the OC tracer was specified to be 0. All OC absorption was due to the BrC tracer."

5. Page 10, section 5.3, equations (6) and (7): I think that the parameterizations in Sahel et al., (2014) are for absorption spectral dependence of the total OA including both absorbing and non-absorbing components. But here they are applied to the absorbing components only?
**Author's Response:** Equations (6) and (7) are the functions for the wavelength dependence of imaginary refractive index for all kinds of particles, they obey a power relationship with the wavelength ratio and an exponent w. The parameterizations in Saleh et al., (2014), described in Eq. 4, shows a relationship between BC/OA ratio and the k of total OA including both absorbing and non-absorbing components. We convert $k_{OA}$ to $k_{BrC}$ using Eq. 3 and apply it to the absorbing components.

6. Page 10, line 311: please clarify what two sets of radiative fluxes are used to calculate the BrC DRE
**Author's Response**: We simulated the clear sky net solar flux at the top of atmosphere with all aerosols (FSNTOAC, the BrC tracers have only absorption and no scattering) and the clear sky net solar flux at the top of atmosphere with all aerosols but no BrC absorption (FSNTOAC_noBrC, the BrC tracers have no absorption or scattering). We calculated BrC DRE by subtracting FSNTOAC and FSNTOAC_noBrC. Lines 366-368 is updated with this information.

Brown, H., Liu, X., Feng, Y., Jiang, Y., Wu, M., Lu, Z., Wu, C., Murphy, S. and Pokhrel, R., 2018. Radiative effect and climate impacts of brown carbon with the Community Atmosphere Model (CAM5). Atmospheric Chemistry and Physics, 18, 17745-17768, https://doi.org/10.5194/acp-18-17745-2018, 2018.

Browne, E. C., Zhang, X., Franklin, J. P., Ridley, K. J., Kirchstetter, T. W., Wilson, K. R., Cappa, C. D., and Kroll, J. H.: Effect of heterogeneous oxidative aging on light absorption by biomass-burning organic aerosol, Aerosol Science and Technology, 1-15, https://doi.org/10.1080/02786826.2019.1599321, 2019.

Forrister, H., Liu, J., Scheuer, E., Dibb, J., Ziemba, L., Thornhill, K. L., Anderson, B., Diskin, G., Perring, A. E., and Schwarz, J. P.: Evolution of brown carbon in wildfire plumes, Geophysical Research Letters, 42, 4623-4630, https://doi.org/10.1002/2015GL063897, 2015.

Jo, D. S., Park, R. J., Lee, S., Kim, S.-W., and Zhang, X.: A global simulation of brown carbon: implications for photochemistry and direct radiative effect, Atmospheric Chemistry and Physics, 16, 3413-3432, https://doi.org/10.5194/acp-16-3413-2016, 2016.

Kirchstetter, T., and Thatcher, T.: Contribution of organic carbon to wood smoke particulate matter absorption of solar radiation, Atmospheric Chemistry and Physics, 12, 6067-6072, https://doi.org/10.5194/acp-12-6067-2012, 2012.

Liu, J., Bergin, M., Guo, H., King, L., Kotra, N., Edgerton, E., and Weber, R.: Size-resolved measurements of brown carbon in water and methanol extracts and estimates of their contribution to ambient fine-particle light absorption, Atmospheric Chemistry and Physics, 13, 12389-12404, https://doi.org/10.5194/acp-13-12389-2013, 2013.

McMeeking, G. R.: The optical, chemical, and physical properties of aerosols and gases emitted by the laboratory combustion of wildland fuels, Colorado State University, 2008.

Nakayama, T., Matsumi, Y., Sato, K., Imamura, T., Yamazaki, A., and Uchiyama, A.: Laboratory studies on optical properties of secondary organic aerosols generated during the photooxidation of toluene and the ozonolysis of α-pinene, Journal of Geophysical Research: Atmospheres, 115, https://doi.org/10.1029/2010JD014387, 2010.

Saleh, R., Robinson, E. S., Tkacik, D. S., Ahern, A. T., Liu, S., Aiken, A. C., Sullivan, R. C., Presto, A. A., Dubey, M. K., and Yokelson, R. J.: Brownness of organics in aerosols from biomass burning linked to their black carbon content, Nature Geoscience, 7, 647-650, https://doi.org/10.1038/ngeo2220, 2014.

Wang, X., Heald, C. L., Liu, J., Weber, R. J., Campuzano-Jost, P., Jimenez, J. L., Schwarz, J. P., and Perring, A. E.: Exploring the observational constraints on the simulation of brown carbon, Atmospheric Chemistry and Physics, 18, 635, https://doi.org/10.5194/acp-18-635-2018, 2018.

Zhang, Y., Forrister, H., Liu, J., Dibb, J., Anderson, B., Schwarz, J. P., Perring, A. E., Jimenez, J. L., Campuzano-Jost, P., and Wang, Y.: Top-of-atmosphere radiative forcing affected by brown carbon in the upper troposphere, Nature Geoscience, 10, 486, https://doi.org/10.1038/ngeo2960, 2017.

---

## Author Comment (AC6) · 20 Sep 2019

[Figure]

**Figure S1**. Simulated (a) and MODIS observed (b) 550 nm AOD data averaged for the months and regions in which fire emissions account for < 50% of the total AOD for 2010. AERONET measurements in the corresponding months and regions are shown as color-coded open circles in (a). MODIS data in the shaded Arctic region in (b) are not used due to the uncertainty of MODIS retrieval above bright surface (Remer et al., 2013).

[Figure]

**Figure S2.** Comparison between observed and simulated vertical profiles of BrC absorption at 365 nm. Black lines and shaded areas show the means and standard deviations of the observations binned in 1-km intervals, respectively. The colored vertical lines and horizontal bars show the means and standard deviations of corresponding model results, respectively. Model sensitivity simulations of BrC are listed in Table 2.

**Table S1a. RRTMG wavelength boundaries for shortwave**

| Band Index | Lower boundary Wavelength (nm) | Upper Boundary Wavelength (nm) |
|---|---|---|
| 1 | 3077 | 3846 |
| 2 | 2500 | 3077 |
| 3 | 2150 | 2500 |
| 4 | 1942 | 2150 |

| | | |
|---|---|---|
| 5 | 1626 | 1942 |
| 6 | 1299 | 1626 |
| 7 | 1242 | 1299 |
| 8 | 778 | 1242 |
| 9 | 625 | 778 |
| 10 | 442 | 625 |
| 11 | 345 | 442 |
| 12 | 263 | 345 |
| 13 | 200 | 263 |

20

25

30

35

**Table S1b. RRTMG wavelength boundaries for longwave**

| Band Index | Lower boundary Wavelength (µm) | Upper Boundary Wavelength (µm) |
|---|---|---|
| 1 | 28.57 | 1000 |
| 2 | 20 | 28.57 |
| 3 | 15.87 | 20 |
| 4 | 14.29 | 15.87 |
| 5 | 12.2 | 14.29 |
| 6 | 10.2 | 12.2 |
| 7 | 9.26 | 10.2 |
| 8 | 8.47 | 9.26 |
| 9 | 7.19 | 8.47 |
| 10 | 6.76 | 7.19 |
| 11 | 5.56 | 6.76 |
| 12 | 4.81 | 5.56 |
| 13 | 4.44 | 4.81 |
| 14 | 4.2 | 4.44 |
| 15 | 3.85 | 4.2 |
| 16 | 3.08 | 3.85 |

40

**Reference**

Remer, L., Mattoo, S., Levy, R., and Munchak, L.: MODIS 3 km aerosol product: algorithm and global perspective, Atmospheric Measurement Techniques, 6, 1829, 2013.

50

---

## Referee Report (RR1)

Review of "Modeling global radiative effect of brown carbon…"

For my first major comment, the authors added clarification in Section 3. However, it is important to point out that their definition of BrC is different from the common definition in literature, which is a class of organic aerosols with both scattering and absorption (not only different from Brown et al., 2018). Especially the concept of absorption-only-no-scattering BrC is inconsistent with the experimental perspective. In order to avoid confusion, I suggest to state this difference also in abstract and conclusion, by clarifying that the developed module is for treating BrC absorption properties only. The added sentence "…the DRE from BrC is for light absorption only such that it can be directly compared to BC" seems to imply that BC is absorption only, which is not true. Please revise it.

For my second major comment, they added AOD comparison, but didn't address "…lacks of evaluation of the modeled aerosol absorption". In particular, " There is no model evaluation of aerosol absorption over the tropics where the BrC heating effect is suggested to be important. It would be useful to know how the modeled aerosol total and spectrally-dependent absorption compare with observations, i.e., AERONET data, with vs without BrC parameterizations. This would provide observational constraints for the calculated heating effects due to BrC and BC. "

For my third major comment, the response is acceptable.

All the minor comments are addressed.

---

## Referee Report (RR2)

Review of "Modeling global radiative effect of brown carbon…"

The authors addressed my comments in the second round. For the second comment, they added Figure 6 (c) for AAOD comparison with AERONET. It needs a few clarifications.

First, what wavelength the AAOD is compared at? It makes a difference as BrC absorbs only moderately in 550nm. If the purpose is to show BrC effect, they should compare AAOD at shorter wavelength or spectral dependence. If the purpose is to show the modeled aerosol absorption compares with obs, it may be good to compare at 550nm. Nevertheless, this needs to be clarified. In the conclusion, these findings from the comparison of modeled aerosol absorption with AERONET should be included in the discussion of estimated BrC DRE vs BC DRE, i.e., could the underestimated AAOD (if it's at 550nm) imply that the relative importance of BrC DRE to BC DRE is overestimated in the current model?

Second, why not use least-square-root regression for AAOD, as it is for AOD? It is not explained why a different regression is needed for AAOD comparison. Does the inclusion of BrC improve the mean AAOD? It seems from Figure 6c still low-biased. At least the mean statistics should be calculated and presented.

lastly, the sentence "Similarly, we used the same criterion for model data" needs clarification: is the model AOD at 440nm >= 0.4 threshold applied to filter the model AAOD at the monthly mean or daily or hourly time intervals? The CAM AOD is obviously low-biased compared with AERONET: does it make sense to use the same AOD threshold at 440nm>0.4 to select the model data?

---

## Author Response (AR2)

**Author's response to referee #1**

This paper continues to have major flaws of presentation and interpretation. The wording and presentation of results are somewhat careless; there are also grammatical errors throughout. The manuscript is not publishable in current form.

**Author's response**: The manuscript has been revised to address these issues.

1. Section 4.1 and Figure 5: There is no quantitative evaluation of BC mass concentrations presented here. Given that the conclusion on relative heating rates rely on an accurate simulation of BC (as well as BrC), the authors should provide an exact comparison (e.g. mean vertical profiles of BC for each HIPPO campaign comparing model and observations as in Figure 7).

**Author's response**: Improvements with the model have been shown in Figure 4. We compare here the mean vertical profiles of BC from observations with the model for each HIPPO campaign, shown in Figure R1 below. Compared to the default CAM5 simulation, the modified CAM5 results agree better with the observation in HIPPO 1, 2 and 3. The modification overestimates BC during HIPPO 4 and 5, but better simulated the shape of the profile compared to the default CAM 5. However, we did not include the results in the paper because HIPPO missions covered from the North to South Pole and averaging over the vast latitude range provides little useful information on model performance. Figure 4 provided much more details than Figure R1 in our opinion.

[Figure]

**Figure R1.** Comparison of the BC vertical profiles during HIPPO missions. Black lines and shaded areas show the means and standard deviations of the observations binned in 1-km intervals, respectively. The colored vertical lines and horizontal bars show the means and standard deviations of the default CAM5 results (blue) and the modified CAM5 results (red).

We understand that the reviewer had difficulty quantifying the improvements in our model simulations. We therefore included vertical profile comparisons for 5 latitude bins (90°S-60°S,

60°S-20°S, 20°S-20°N, 20°N-60°N, 60°N-90°N) in Figure R2, which is included in the paper as the new Figure 5. For all latitude bins, the modified CAM5 simulations agree better with the observation than the default CAM5. However, the modified CAM5 simulations still overestimate BC over the tropical middle and upper troposphere. This overestimation may potentially lead to an underestimation of BrC to BC ratio over the remote tropics.

In the manuscript, we added this figure as Figure 5, and describe this figure in Lines 252-256: "*Figure 5 shows the comparison between BC vertical profiles during all HIPPO campaigns with CAM5 simulations for 5 latitude bins (90°S-60°S, 60°S-20°S, 20°S-20°N, 20°N-60°N, 60°N-90°N), respectively. The modified CAM5 simulations agreed better with the observations in all regions, but still overestimated BC in the middle and upper troposphere over the tropics, which may lead to a low bias in the model simulated BrC/BC heating ratio in the tropics (to be discussed in section 5.3).*"

In Lines 496-499, we noted the model uncertainty from the inaccuracy of BC constrains: "*The uncertainty of model simulated BC also affects the comparison between the DRE and heating of BC and BrC. For example, the model overestimates of BC in the middle and upper tropical troposphere (Figure 5) may lead to an underestimate of the BrC to BC DRE ratio over the remote tropics.*"

[Figure]

**Figure R2**. Comparison of observed and simulated BC vertical profiles during HIPPO missions for the latitude bins of 90° S-60° S, 60° S-20°S, 20° S-20° N, 20° N-60°N and 60° N-90°N. Black lines and shaded areas show the means and standard deviations of the observations binned in 1-km intervals, respectively. The colored vertical lines and horizontal bars show the means and standard deviations of the default (blue) and modified CAM5 results (red), respectively.

2. Section 4.1: Why do the authors choose to compare to AOD (which as they rightly point out is dominated by aerosol scattering) rather than AAOD estimates from AERONET or OMI UV AI? There does not appear to be much value to the comparisons presented here.

**Author's response**: Thank you for your suggestions. We now added the comparison to AERONET AAOD observations from 2005 to 2014. Figure R3 is included in the paper as new Figure 6(c).

[Figure]

**Figure R3**. Comparison of AAOD between CAM and AERONET from 2005 to 2014 with BrC (red) and without BrC (blue). The solid lines denote PC regression lines for model results with and without BrC absorption, and the corresponding regression slope (*k*) values are shown. The dashed line denotes the 1:1 reference line.

We revised the title of Section 4.2 to "*Aerosol optical depth (AOD) and absorption aerosol optical depth (AAOD) over fire emission dominated regions*". For the description of this figure, we added in Section 4.2, Lines 279-289: *"In addition, we compared the model simulations to the AAOD data from the AERONET version 3 level 2.0 inversion dataset (Holben et al., 2006). Since the AAOD estimation is highly uncertain in the low AOD conditions (Dubovik et al., 2000), we used only AAOD measurements for AOD at 440 nm ≥ 0.4 (Holben et al., 2006). Similarly, we used the same criterion for model data. Figure 6(c) compares the monthly mean 2005-2014 AERONET AAOD data over fire-dominated regions and months with the corresponding monthly mean model results. The observations showed significant interannual variability, which was not included in the model results for a climatological 2010 year. The AAOD absorption is higher by 8.5% on average when BrC is considered in the model simulations. We performed principal-component regression analysis of observed and simulated data. With BrC absorption, the simulated higher AAOD data are in better agreement with AERONET observations with a regression slope of 0.95 compared to a slope of 0.66 for the simulation without BrC absorption. The remaining offset of ~0.01 in the AAOD is insignificant compared to the variability of the observations."*

3. Section 5.1: This section and discussion lack logical flow and the wording is often inconsistent with the figures. From Figure 7, the baseline simulation (NCNB) appears to perform equally well to the "best" simulation (ICB) which includes photobleaching and

reduced scavenging. The authors therefore need to present a more compelling justification for their "best" simulation. This should begin with what (if any) flaws are apparent from the baseline simulation and then carefully explore, individually, the effects of photobleaching and reduced scavenging. And then finally make a case for their "best" simulation.

a. The comparison of Figure S2 should be moved into the main text, along with a comparison of OA mass concentrations. The authors need to show that their model simulation simultaneously meets the constraints of mass and optics for both BC and OA/BrC, and to make their case that the ICB simulation is superior to the NCNB simulation in this regard.

**Author's response:** We believe the reviewer is assuming that we need to test a flawed model, like NCNB, with the observations. This is not true. Numerous papers have shown that photobleaching occurs in the atmosphere. Considering NCNB as a "base" model does not correspond to our understanding of BrC at present. Therefore, we did not go into the details in the previous revision. We again address this reviewer comment.

We previously responded to the reviewer's question by pointing out the overestimation of BrC at 0-8 km in the NCNB simulation. To make it apparent, we moved the original supplement Figure S2 into the main paper as a panel of Figure 7 (now Figure 8 in the revised manuscript). We revised the discussion of the model evaluation with DC3 and SEAC[4]RS observations. The following paragraph was added in Lines 320-335.

*"Table 2 lists all sensitivity simulations. For BrC and BrC/BC simulations, Figure 8 shows that the NCNB model clearly overestimated BrC compared to the observations at 0-8 km (the overestimate is not as apparent in the BrC/BC comparison because it is a logarithmic scale). The overestimation reflected the importance of photo-bleaching (Forrister et al., 2015; Sareen et al., 2013; Lee et al., 2014; Wang et al., 2016; Wong et al., 2017; Wong et al, 2019; Zhong and Jang, 2011). The overestimation in the lower troposphere in NCNB led to a reasonable simulation of BrC in the upper troposphere, although the underestimation at 12 km was obvious relative to the ICB simulation during the DC3 experiment. Similarly, considering enhanced convective transport, but not photo-bleaching, the ICNB simulation clearly overestimated BrC absorption relative to the observations. Including photo-bleaching, but not enhanced convective transport of BrC, the NCB simulation clearly underestimated BrC and the BrC/BC ratio in comparison to the observations. We also included a simulation of ICBB, in which enhanced convective transport of BrC was included with photo-bleaching. Compared to the observations, upper tropospheric BrC and the BrC/BC ratio in the ICBB simulation were clearly underestimated. At 12 km, the observed BrC/BC ratio is ~10 and ~20 times higher than BrC/BC near the surface during DC-3 and SEAC4RS, respectively. This increase in the BrC/BC ratio in the upper troposphere was captured by the ICB simulation. On the basis of our current understanding of BrC processes (Forrister et al., 2015; Sareen et al., 2013; Lee et al., 2014; Wang et al., 2016; Wong et al., 2017; Wong et al, 2019; Zhang et al., 2017; Zhong and Jang, 2011) and the model evaluation with the observations, we chose the ICB simulation to investigate the effects of global BrC radiative forcing."*

As to the reviewer's suggestion of looking into the OA/BrC ratio, we do not see a reason why it is necessary. Our BrC simulation (or the BrC tracer) only simulated organic absorption, not scattering. So the model OA simulation is not affected by the BrC simulation at all. The reviewer misunderstood what was done in the model.

We clarified in Section 3, Lines 188-192: "*However, it should be noted that BrC is a class of organic aerosols that both scatter and absorb light. We analyzed in this study the effect of*

*BrC light absorption. The model simulation of OC mass and scattering was not affected by the simulation of a BrC tracer that only absorbs light. In the following analysis, the DRE from BrC is for light absorption only such that it represents the DRE of the OC absorption and can be compared to the DRE of the BC absorption..*"

Section 6, Lines 474-475: "*We developed a module for simulating the effects of brown carbon light absorption in CESM CAM5 and conducted two sets of model experiments, 2010 with nudged meteorological fields and 5-year free-running simulations*."

b. Lines 306-307: This statement is incorrect, the NCNB and ICB simulations are very similar.

**Author's response:** Please see our previous response. The two simulations of BrC are very different at 0-8 km.

c. Lines 308: The BrC/BC ratio is similarly underestimated in the ICB simulation > 8km

**Author's response:** We revised this sentence (now in Lines 323-325) to: "*The overestimation in the lower troposphere in NCNB led to a reasonable simulation of BrC in the upper troposphere, although the underestimation at 12 km was obvious relative to the ICB simulation during the DC3 experiment.*" Please also see our previous response.

d. Line 310: This sentence is incorrect. The ICNB and NCNB simulations do not show very different profiles (i.e. lower trop to upper trop ratios) in Figure 7. The second pairing "ICB&NCB" is the wrong pair of simulations to compare – the NCB simulation should be compared to the ICBB simulation

**Author's response:** We do not understand what the reviewer referred to. The difference between ICNB and NCNB is very obvious (please see Figure 8). The paragraph is rewritten. Please see previous responses.

e. Lines 315-316: If photobleaching were implemented as in previous studies (with a 25% floor) would the BrC/BC ratios in the NCB match the observations?

**Author's response:** We applied a 6% floor in this study based on the field measurements of Forrister et al. (2015). If valid measurement studies can be ignored arbitrarily, we do not see the point of doing model simulation or evaluation in the first place.

f. Lines 319-320: These statements are false. Figure S2 shows that only the ICNB simulation clearly overestimates BrC in the UT for both campaigns. Figure 7 shows that only two simulations (ICBB and NCB) clearly underestimate FT-UT BrC/BC

**Author's response:** Again, we do not understand how the reviewer missed the obvious differences among the model simulations. The section was revised. Please see the previous responses.

g. Lines 326-327: This statement is not supported; the authors have not shown the reader that the ICB simulation is superior to the baseline.

**Author's response:** If by baseline, the reviewer meant the NCNB model, we have responded to this comment. The NCNB cannot be used to simulate BrC absorption at this time for many good reasons as we discussed. We strongly disagree with the reasoning of this comment.

4. Figure 9c: requires a better choice of color bar so that the value of 1 can be distinguished. For current figure it is not clear where BC heating rates exceed BrC and vice versa.

**Author's response:** The following figure is updated as Figure 9 (now Figure 10). We added a

sentence in the caption of Figure 10: "*In (c), BrC/BC DRE ratios larger than 1.0 are specified by a different color bar.*"

[Figure]

5. Lines 401-408: The authors should be quantitative here to clarify the relative heating rates. From Figure 13 for > 600 hPA the BrC/BC mean heating rate ratio ranges from 0.9-1.2.

**Author's response:** Quantitative analysis of the heating rates have been added in Section 5.4, Lines 437-439: "*Globally, the average BrC/BC heating rate ratio is 15% below 500 hPa and is 44% above 500 hPa. In deep convection regions, the average BrC/BC heating rate ratio is 60% below 500 hPa and 118% above 500 hPa, indicating in deep convection regions, atmospheric heating of BrC is stronger than that of BC.*"

Minor

1. Line 29-30, suggest temper the language given uncertainties "This suggests that the

contribution of BrC heating to the Hadley circulation and latitudinal expansion of the tropics may be comparable to BC heating."

**Author's response:** We changed the wording.

2. Lines 98-100: indicate explicitly whether the CAM5 meteorology was nudged to the same meteorological year as the observations (i.e. 2009-2011 for HIPPO, 2013 for SEAC4RS and DC3).

**Author's response:** We revised this sentence in Lines 98-101 to "*For the simulations used to compare with field observations, we nudged CAM5 meteorological field (temperature, humidity, wind, surface pressure and heat) to the same meteorological year, month and day as the observations using GEOS-5.2.0 meteorological data products (Suarez et al., 2008) every 6 hours in order to evaluate the model simulations with BrC observations (Ma et al., 2013; Chipperfield, 2006).*"

3. Lines 133-135: how do these optical properties compare with those used in previous modeling studies? And the literature more generally?

**Author's response:** We expanded this paragraph on comparing to previous studies in Lines 134-136: "*The MAE values are lower than the estimation by Bond et al. (2013) (11 $m^2/g$) and Jacobson (2012) (16 $m^2/g$ including high-RH conditions), and are higher than the estimation by Schulz et al. (2006) (7.9±1.9 $m^2/g$).*" We compared the optical properties of BrC with previous studies in Sections 3.1 and 3.2.

4. Lines 283-285: the correlation coefficient (R) is obtained for a population of points, how is it used here as a point-by-point filter?

**Author's response:** The correlation between CO and $CH_3CN$ is calculated within fire plumes, which have a population of points. According to de Gouw et al. (2004) and Liu et al. (2014), if the coefficient of determination ($r^2$) between CO and CH3CN was higher than 0.5 during the period of enhanced CO, the plume was designated as biomass burning. We revised the description (now in Lines 298-300) to: "*Fresh fire plume data, diagnosed by plumes with a coefficient of determination between CO and $CH_3CN$ > 0.5 during the period of enhanced CO, were not included in model evaluation as in previous studies (De Gouw et al., 2004; Liu et al., 2014).*"

5. Section 5.1: missing information and references for measurements from SEAC4RS and DC3 used in this study.

**Author's response:** We revised the description of SEAC4RS and DC3 in Lines 293-297: "*We evaluated BrC model simulations using the measured BrC absorption data from the airborne measurements of Studies of Emissions, Atmospheric Composition, Clouds and Climate Coupling by Regional Surveys (SEAC4RS) (Toon et al., 2016) and Deep Convective Clouds and Chemistry Project (DC3) field experiments (Barth et al., 2015). The SEAC4RS campaign was conducted during August 6 to September 23, 2013 over the central and southeast U.S., and the DC3 campaign was conducted from May 18 to June 22, 2012 over a similar region.*"

6. Lines 299-300: can the authors use a biomass burning tracer such as acetonitrile or HCN to confirm that this BC is from non-fire sources?

**Author's response:** Thank you for the suggestion. We corrected the error in Lines 314-317: "*The underestimation at 2-5 km during SEAC$^4$RS likely reflects underestimated fire emissions since the coefficient of determination ($R^2$) is 0.6 for HCN and BC at 2-5 km and it is 0.5 for HCN and BrC, reflecting the effects of biomass burning emissions on BC and BrC.*"

**Author's response to referee #2**

For my first major comment, the authors added clarification in Section 3. However, it is important to point out that their definition of BrC is different from the common definition in literature, which is a class of organic aerosols with both scattering and absorption (not only different from Brown et al., 2018). Especially the concept of absorption-only-no-scattering BrC is inconsistent with the experimental perspective. In order to avoid confusion, I suggest to state this difference also in abstract and conclusion, by clarifying that the developed module is for treating BrC absorption properties only. The added sentence "…the DRE from BrC is for light absorption only such that it can be directly compared to BC" seems to imply that BC is absorption only, which is not true. Please revise it.

**Author's Response:** Thank you for your comments. To avoid the confusion about our treatment of BrC in the model, we revised the abstract in Lines 16-21 to "*In this study, we derived a BrC global biomass burning emission inventory on the basis of the Global Fire Emissions Database 4 (GFED4), developed a module to simulate the light absorption of BrC in the Community Atmosphere Model version 5 (CAM5) of Community Earth System Model (CESM) model, and investigated the photo-bleaching effect and convective transport of BrC on the basis of Studies of Emissions, Atmospheric Composition, Clouds and Climate Coupling by Regional Surveys (SEAC4RS) and Deep Convective Clouds and Chemistry Project (DC3) measurements.*" We also revised Section 3, Lines 188-192 to "*However, it should be noted that BrC is a class of organic aerosols that both scatter and absorb light. We analyzed in this study the effect of BrC light absorption. The model simulation of OC mass and scattering was not affected by the simulation of a BrC tracer that only absorbs light. In the following analysis, the DRE from BrC is for light absorption only such that it represents the DRE of the OC absorption and can be compared to the DRE of the BC absorption.*", so that our treatment of BrC is clarified and the description of BC DRE is not misleading. We also revised the conclusion in Lines 474-475 to: "*We developed a module for simulating the effects of brown carbon light absorption in CESM CAM5 and conducted two sets of model experiments, 2010 with nudged meteorological fields and 5-year free-running simulations.*"

For my second major comment, they added AOD comparison, but didn't address "…lacks of evaluation of the modeled aerosol absorption". In particular, " There is no model evaluation of aerosol absorption over the tropics where the BrC heating effect is suggested to be important. It would be useful to know how the modeled aerosol total and spectrally-dependent absorption compare with observations, i.e., AERONET data, with vs without BrC parameterizations. This would provide observational constraints for the calculated heating effects due to BrC and BC. "

**Author's Response:** Recent measurements from the Atmospheric Tomography Mission (ATom) provides BrC measurements in remote regions, including the tropics. The Atom measurements show a BrC/BC absorption ratio reaching ~1 in the upper troposphere over the remote tropics. A manuscript describing the measurements and analysis is in preparation. In addition, we compared the model results of the absorption aerosol optical depth (AAOD) to AERONET, and the figure was added in the main text (Figure 6(c), shown below).

[Figure]

**Figure 6(c)**. Comparison of AAOD between CAM and AERONET from 2005 to 2014 with BrC (red) and without BrC (blue). The solid lines denote PC regression lines for model results with and without BrC absorption, and the corresponding regression slope (*k*) values are shown. The dashed line denotes the 1:1 reference line.

We revised the title of Section 4.2 to "*Aerosol optical depth (AOD) and absorption aerosol optical depth (AAOD) over fire emission dominated regions*". For the description of this figure, we added in Section 4.2, Lines 279-289: "*In addition, we compared the model simulations to the AAOD data from the AERONET version 3 level 2.0 inversion dataset (Holben et al., 2006). Since the AAOD estimation is highly uncertain in the low AOD conditions (Dubovik et al., 2000), we used only AAOD measurements for AOD at 440 nm ≥ 0.4 (Holben et al., 2006). Similarly, we used the same criterion for model data. Figure 6(c) compares the monthly mean 2005-2014 AERONET AAOD data over fire-dominated regions and months with the corresponding monthly mean model results. The observations showed significant interannual variability, which was not included in the model results for a climatological 2010 year. The AAOD absorption is higher by 8.5% on average when BrC is considered in the model simulations. We performed principal-component regression analysis of observed and simulated data. With BrC absorption, the simulated higher AAOD data are in better agreement with AERONET observations with a regression slope of 0.95 compared to a slope of 0.66 for the simulation without BrC absorption. The remaining offset of ~0.01 in the AAOD is insignificant compared to the variability of the observations.*"

For my third major comment, the response is acceptable.
All the minor comments are addressed.
**Author's Response:** Thank you for your comments.

[revised manuscript text omitted]

**5 Results**

**5.1 Model simulations of BrC for DC3 and SEAC[4]RS missions**

We evaluated BrC model simulations using the measured BrC absorption data from the airborne measurements of Studies of Emissions, Atmospheric Composition, Clouds and Climate Coupling by Regional Surveys (SEAC[4]RS) (Toon et al., 2016) and Deep Convective Clouds and Chemistry Project (DC3) field experiments (Barth et al., 2015). The SEAC[4]RS campaign was conducted during August 6 to September 23, ; Liu et al., 2013 over the central and southeast U.S., and the DC3 campaign was conducted from May 18 to June 22, 2012 over a similar region.; Zhang et al., 2017). Flight tracks for these experiments were shown in Figure 76. Fresh fire plume data, diagnosed by plumesdata points with a correlation coefficient of determination between CO and CH₃CN > 0.5 during the period of enhanced CO, were not included in model evaluation as in previous studies (De Gouw et al., 2004; Liu et al., 2014).

We described in section 3.1 the rationale for sensitivity simulations to evaluate the effects of BrC photo-bleaching and convective wet scavenging. The model sensitivity simulations are listed in Table 2. In the NCNB (base) model, neither effect was included. In the NCB model, the photo-bleaching effect is included. In the ICNB model, the wet scavenging efficiency of convective transported BrC was decreased from 75% simulated in the base model to 30%, such that ~70% of BrC was transported through convection to the free troposphere as suggested by Zhang et al. (2016). In the ICB model, both photo-bleaching and reduced convective scavenging effects were included. The ICBB model is similar to ICB model, but photo-bleaching of all BrC was included; in the other models including the photo-bleaching effect, only non-convectively transported BrC was affected (Zhang et al., 2017).

Figure 87 shows the observed vertical profiles of BrC absorption, BrC to BC absorption ratio (BrC/BC ratio) and concentrations of BC and CO during the DC3 and SEAC[4]RS experiments in comparison to the corresponding model simulation results. The difference between BC and CO vertical profiles is negligible among the sensitivity simulations. Simulated mean BC concentrations are within the uncertainties of the measurements. The underestimation at 2-5 km during SEAC[4]RS likely reflects underestimated suggests transport of non-fire emissionsBC to this region since the coefficient of determination ($R^2$)observed enhancement is 0.6insignificant 
[revised manuscript text omitted]

---

## Author Response (AR3)

**Author's response to referee #1**

The authors addressed my comments in the second round. For the second comment, they added Figure 6 (c) for AAOD comparison with AERONET. It needs a few clarifications.

**Author's response**: Thank you for the suggestions. The issues have been clarified in the revised manuscript.

First, what wavelength the AAOD is compared at? It makes a difference as BrC absorbs only moderately in 550nm. If the purpose is to show BrC effect, they should compare AAOD at shorter wavelength or spectral dependence. If the purpose is to show the modeled aerosol absorption compares with obs, it may be good to compare at 550nm. Nevertheless, this needs to be clarified. In the conclusion, these findings from the comparison of modeled aerosol absorption with AERONET should be included in the discussion of estimated BrC DRE vs BC DRE, i.e., could the underestimated AAOD (if it's at 550nm) imply that the relative importance of BrC DRE to BC DRE is overestimated in the current model?

**Author's response**: AAOD at 550 nm was compared. We clarified it by adding in the manuscript at Lines 288-289 "*To show the performance of the model simulation of aerosol absorption, here we compare the observed and simulated AAOD values at 550 nm, which is near the peak wavelength of solar intensity.*" and Lines 299-300 "*Globally, the AAOD absorption at 550 nm is higher by 8.5% on average when BrC is considered in the model simulations.*" To indicate the potentially higher AAOD impact at lower wavelengths, we added at Lines 289-291: "*Because of the strong wavelength dependence of BrC absorption, the enhancement of AAOD by BrC absorption at wavelengths lower than 550 nm is more significant.*"

The AAOD comparison does not imply that BrC is underestimated since the simulated AAOD is lower than the observations. We added a clarification at Lines 297-299 "*For these observations, the model underestimated the AERONET AAOD observations by 39% without BrC absorption. Including BrC absorption reduced the low bias to 17%, which is well within the large variability of the observations.*"

In this year's AGU fall meeting, we presented model evaluations with the ATom global observations. Model simulated BrC/BC ratios are lower than the observations. So we have no reason to suspect that BrC in the model simulations was overestimated. If anything, BrC is underestimated. The ATom evaluation results will be reported as part of an ATom BrC paper.

Second, why not use least-square-root regression for AAOD, as it is for AOD? It is not explained why a different regression is needed for AAOD comparison. Does the inclusion of BrC improve the mean AAOD? It seems from Figure 6c still low-biased. At least the mean statistics should be calculated and presented.

**Author's response**: The inclusion of BrC increased the mean AAOD by 8.5%. We stated previously, now at Lines 299-300: "*Globally, the AAOD absorption at 550 nm is higher by 8.5% on average when BrC is considered in the model simulations.*" We added in the text at Lines

297-299 "*For these observations, the model underestimated the AERONET AAOD observations by 39% without BrC absorption. Including BrC absorption reduced the low bias to 17%, which is well within the large variability of the observations*."

We should have used PC regression in all analysis. Thank you for pointing it out. PC regression analysis treats the deviations from the regression line equally for x and y variables. In comparison, the least-squares region only minimizes the deviations in y axis from the regression line. For model-observation comparison, PC regression is therefore more appropriate. The regression lines are very similar. In the revised manuscript, we updated Figure 6 (a) and (b) and updated corresponding description of the regression for AOD.

lastly, the sentence "Similarly, we used the same criterion for model data" needs clarification: is the model AOD at 440nm >= 0.4 threshold applied to filter the model AAOD at the monthly mean or daily or hourly time intervals? The CAM AOD is obviously low-biased compared with AERONET: does it make sense to use the same AOD threshold at 440nm>0.4 to select the model data?

**Author's response**: Yes, we agree with the reviewer. We clarified the description at Lines 283-288: "*Monthly mean AAOD data were computed for AERONET sites with more than 10 days of daily averaged observed AOD at 440 nm > 0.4 in a month. Because of the model underestimation, the corresponding model threshold of AOD at 440 nm is 0.315 based on the PC regression between AERONET observations and model simulation results (Figure 6(a)). Daily model results with AOD at 440 nm > 0.315 were used to compute simulated monthly means for the grid cells corresponding to the AERONET sites*." Figure 6(c) was updated based on this filtered model result, the updated Figure 6 is shown below. The regression statistic changes are updated in the paper.

[revised manuscript text omitted]